# Design-Based Bandits Under Network Interference: Trade-Off Between Regret and Statistical Inference

**Zichen Wang[1]*  Haoyang Hong[2]*  Chuanhao Li[3]  Haoxuan Li[4]  Zhiheng Zhang[5,6]†**

**Huazheng Wang[2]**

[1]Department of ECE and CSL, UIUC
[2]School of EECS, Oregon State University
[3]Department of Industrial Engineering, Tsinghua University
[4]Center for Data Science, Peking University
[5]School of Statistics and Data Science, Shanghai University of Finance and Economics,
Shanghai 200433, P.R. China
[6]Institute of Data Science and Statistics, Shanghai University of Finance and Economics,
Shanghai 200433, P.R. China

## Abstract

In multi-armed bandits with network interference (MABNI), the action taken by one node can influence the rewards of others, creating complex interdependence. While existing research on MABNI largely concentrates on minimizing regret, it often overlooks the crucial concern that an excessive emphasis on the optimal arm can undermine the inference accuracy for sub-optimal arms. Although initial efforts have been made to address this trade-off in single-unit scenarios, these challenges have become more pronounced in the context of MABNI. In this paper, we establish, for the first time, a theoretical Pareto frontier characterizing the trade-off between regret minimization and inference accuracy in adversarial (design-based) MABNI. We further introduce an anytime-valid asymptotic confidence sequence along with a corresponding algorithm, `EXP3-N-CS`, specifically designed to balance the trade-off between regret minimization and inference accuracy in this setting.

## 1 Introduction

Network interference [Leung, 2022a,b, 2023, Imbens, 2024], a well-known concept in causal inference, describes a phenomenon where the treatment assigned to one individual can influence the outcomes of others. It has been extensively studied across various disciplines, with significant applications in economics [Arpino and Mattei, 2016, Munro et al., 2021] and the social sciences [Bandiera et al., 2009, Bond et al., 2012, Paluck et al., 2016, Imbens, 2024]. Due to its broad real-world relevance, this concept in causal inference has recently been explored and recognized by researchers in online learning. Consequently, it has begun to be frequently applied in multi-armed bandits [Agarwal et al., 2024, Jia et al., 2024, Zhang and Wang, 2024].

To effectively identify causal effects under network interference, a common approach involves conducting randomized experiments to estimate causal effects from experimental data [Leung, 2022a,b, 2023, Gao and Ding, 2023]. Specifically, researchers design estimators that leverage feedback collected from each individual (commonly referred to as *potential outcomes* in the causal inference

---

*Equal contribution.

†Corresponding author. Email: `zhangzhiheng@mail.shufe.edu.cn`.

39th Conference on Neural Information Processing Systems (NeurIPS 2025).

literature). They primarily focus on ensuring unbiasedness and controlling the variance of these estimators. However, in practice, such experiments are often conducted over multiple rounds, introducing a dynamic aspect to individual feedback. In this setting, the aforementioned *potential outcomes* are also referred to as *rewards* in the online learning literature, as they contribute to cumulative regret, which quantifies the overall welfare loss incurred throughout the experiment [Simchi-Levi and Wang, 2024]. Once the experiment concludes, data collected in earlier rounds can be utilized to improve social welfare in future applications [Mok et al., 2021]. For instance, when evaluating the effectiveness of different drug treatments, researchers may not only seek to maximize treatment efficacy during the trial but also estimate the relative differences in treatment effects across drugs based on experimental data. This process necessitates a careful balance between optimizing the *estimation accuracy* of causal effects and minimizing the *cumulative regret* incurred during the experiment [Simchi-Levi and Wang, 2024, Zhang and Wang, 2024]. Furthermore, researchers may wish to continuously infer causal effects throughout the experiment, allowing them to make informed decisions about when to stop based on data-driven metrics or predefined thresholds [Ham et al., 2023, Woong Ham et al., 2023, Liang and Bojinov, 2023]. This type of continual inference often requires *anytime-validity*, ensuring that statistical inferences remain robust regardless of the time at which they are made [Lindon and Malek, 2022, Waudby-Smith et al., 2024].

Building on the above observations, three critical learning objectives emerge: (i) conducting continual inference on causal effects, (ii) minimizing cumulative regret, and (iii) designing estimators that leverage collected data to accurately estimate causal effects once the experiment concludes. However, most existing studies fail to address these three objectives simultaneously. For instance, Jia et al. [2024], Agarwal et al. [2024] focus exclusively on regret minimization in MABNI, whereas Ham et al. [2023], Woong Ham et al. [2023] primarily explore continual inference in adversarial MAB using the technique of Asymptotic **C**onfidence **S**equences (CS) [Waudby-Smith et al., 2021]. Similarly, Simchi-Levi and Wang [2024], Zhang and Wang [2024], Duan et al. [2024] investigate the trade-off between regret minimization and causal effect estimation but place little emphasis on continual inference. The most closely related work is Liang and Bojinov [2023], which considers

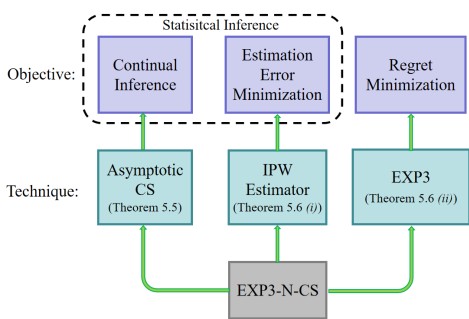

Figure 1: The main contribution of our paper is to study how to achieve these three objectives and to analyze their underlying interrelationships.

all three learning objectives within the framework of adversarial MAB. However, their approach suffers from two key limitations: (i) it does not account for network interference, and (ii) it lacks rigorous theoretical results characterizing the trade-off between causal effect estimation and regret minimization. Building on the above observations, we aim to achieve three key learning objectives in adversarial MABNI and make the following contributions:

- We propose a unified learning framework tailored for the adversarial setting, which we term adversarial `MAB-N`. Furthermore, we establish the first Pareto frontier that delineates the fundamental trade-off between regret and causality-estimation error in adversarial `MAB-N`.

- We develop an anytime-valid asymptotic CS to enable continuous inference in adversarial `MAB-N`. Building on this, we introduce `EXP3-Network-Confidence Sequence` (`EXP3-N-CS`), which integrates our asymptotic CS and is specifically designed to achieve all three learning objectives (i)–(iii).

- We conduct simulation studies to investigate the empirical performance of our `EXP3-N-CS`.

The paper is organized as follows. Section 2 reviews related work. In Section 3, we introduce the core setting of adversarial MABNI, along with the techniques of exposure mapping and clustering, which support the design of `MAB-N`. Section 4 presents our results on Pareto optimality. In Section 5, we introduce the Asymptotic CS technique and our main algorithm, `EXP3-N-CS`. Finally, Section 6 reports the experimental results.

Table 1: The overview of the exploration of these three objectives: Obj. 1 represents regret minimization, Obj. 2 represents continual inference, and Obj. 3 corresponds to minimizing the ATE estimation error.

| Paper | Obj. 1 | Obj. 2 | Obj. 3 | Trade-off | Network | Adversarial |
|---|---|---|---|---|---|---|
| Simchi-Levi and Wang [2024] | ✓ | ✗ | ✓ | ✓ | ✗ | ✗ |
| Woong Ham et al. [2023], Ham et al. [2023] | ✗ | ✓ | ✓ | ✗ | ✗ | ✓ |
| Liang and Bojinov [2023] | ✓ | ✓ | ✓ | ✗ | ✗ | ✓ |
| Jia et al. [2024] | ✓ | ✗ | ✗ | ✗ | ✓ | ✓ |
| Agarwal et al. [2024], Xu et al. [2024] | ✓ | ✗ | ✗ | ✗ | ✓ | ✗ |
| Zhang and Wang [2024] | ✓ | ✗ | ✓ | ✓ | ✓ | ✗ |
| **Ours** | ✓ | ✓ | ✓ | ✓ | ✓ | ✓ |

## 2 Related Work

**Causality inference under network interference.** In the current causality literature, interference is a well-established concept that signifies a violation of the Stable Unit Treatment Value Assumption (SUTVA) [Imbens, 2024]. It arises in scenarios where an individual's treatment potentially influences the outcomes of others, a phenomenon frequently observed in practice. Existing research on offline causal inference under network interference has primarily employed two key methodological approaches: *clustering-based methods* [Zhang and Imai, 2023, Viviano et al., 2023, Zhao, 2024] and *exposure mapping techniques* [Leung, 2022a,b, 2023, Zhao, 2024]. Recently, a growing interest has been in studying the MABNI. For instance, Agarwal et al. [2024] applied Fourier analysis to transform the MABNI problem into a sparse linear stochastic bandit formulation. However, to mitigate the exponential growth of the action space, they imposed a strong sparsity assumption on network structures, restricting the number of neighbors each node can have. In contrast, Jia et al. [2024] explored an MABNI setting without such a sparsity assumption. Their learning framework enforces a switchback design, in which all nodes must adopt the same arm simultaneously. However, this approach does not account for scenarios in which the optimal arm may vary between nodes or subgroups. To address these limitations, Zhang and Wang [2024] proposed a general learning framework, `MAB-N`, which simplifies the stochastic MABNI problem while allowing flexible adjustment of the action space through exposure mapping and clustering techniques. `MAB-N` generalizes the settings considered in Jia et al. [2024] and Agarwal et al. [2024], treating them as special cases (see discussion in Section 3.2). Furthermore, Xu et al. [2024] extended MABNI to the linear contextual bandit setting, incorporating a structured linear relationship between potential outcomes and interference intensity.

**Trade-off between inference and regret.** A substantial body of research has focused on developing statistical methods for inference in stochastic MAB, often deriving statistical tests or central limit theorems while keeping the bandit algorithm largely unchanged [Luedtke and Van Der Laan, 2016, Dimakopoulou et al., 2017, 2019, Zhang et al., 2020a, Dimakopoulou et al., 2021, Hadad et al., 2021, Zhang et al., 2021, Han et al., 2022, Deshpande et al., 2023, Simchi-Levi and Wang, 2024]. These methods enable aggressive regret minimization but are subject to several key limitations: (i) they rely on the SUTVA, (ii) they assume that bandit rewards are i.i.d. samples from specific distribution families, and (iii) they do not support anytime-valid continual inference. Regarding the last limitation, to our knowledge, the only works that attempt continual inference in the adversarial bandit setting are Ham et al. [2023], Woong Ham et al. [2023], Liang and Bojinov [2023]. However, these studies also assume SUTVA and lack a rigorous theoretical analysis of the inference-regret trade-off. To explore this inference-regret trade-off, researchers have first shifted their attention to a simpler problem: balancing estimation accuracy and regret minimization. To our knowledge, the first rigorous trade-off results were provided by Simchi-Levi and Wang [2024] in the stochastic MAB setting, though their approach remains constrained by the SUTVA assumption. Duan et al. [2024] further argues that Pareto optimality—simultaneously achieving optimal regret and estimation accuracy—can be improved by introducing a covariate diversity assumption, provided that there is no network interference. More recently, Zhang and Wang [2024] extended the trade-off results from Simchi-Levi and Wang [2024] to the stochastic `MAB-N` framework, accommodating network interference.

Additional discussions of the related work are provided in the Appendix. The relationship between our work and the most closely related studies is summarized in Table 1.

## 3 Preliminaries

In this section, we first introduce the basic adversarial MABNI framework. Then, we present the techniques of exposure mapping and clustering and outline the adversarial `MAB-N` framework.

### 3.1 Basic framework: adversarial MABNI

We extend the classic single-unit adversarial bandit framework [Auer et al., 2002a] to incorporate network interference [Zhang and Wang, 2024]. Consider a network with $N$ units, represented by the set $\mathcal{U} = \{1, \ldots, N\}$ and the adjacency matrix $\mathbb{H} := \{h_{i,j}\}_{i,j \in [N]} \in \{0,1\}^{N \times N}$ (where $h_{i,j} = 1$ indicates unit $i$ and $j$ are neighbors, while $h_{i,j} = 0$ indicates otherwise). It is worth noting that full knowledge of $\mathbb{H}$ is not strictly required; its necessity depends on the specific design introduced in the following section (see the discussion in Section 3.2). We assume that each unit has a $K$-armed set (action set) denoted as $\mathcal{K} = \{0, 1, 2, \ldots, K - 1\}$. At each round $t$, the learner must assign an arm to each unit, resulting in a super arm (a collection of arms across all units) represented as $A_t = (a_{1,t}, a_{2,t}, \ldots, a_{N,t}) \in \mathcal{K}^{\mathcal{U}}$. Suppose the super arm $A_t$ is pulled in round $t$, the reward derived by unit $i \in \mathcal{U}$ is $Y_{i,t}(A_t) \in [0,1]$, where $Y_{i,t}(\cdot) : \mathcal{K}^{\mathcal{U}} \to \mathbb{R}$ represents the reward function of unit $i$ in round $t$. The terminal time $T$ is not pre-specified and cannot be known to the learner in advance. We define the set of all legitimate design-based bandit instances as $\mathcal{E}_0$, where a legitimate instance $\nu := \{Y_{i,t}(A)\}_{A \in \mathcal{K}^{\mathcal{U}}, i \in \mathcal{U}, t \in [T]}$ satisfies $Y_{i,t}(A) \in [0,1]$ for all $A \in \mathcal{K}^{\mathcal{U}}$, $t \in [T]$ and $i \in \mathcal{U}$.

We aim to design a policy $\pi := (\pi_1, \ldots, \pi_T)$. The $\pi_t$ is a rule that determines the super arm pulled in round $t$ based on the history $\mathcal{H}_{t-1} := \{A_1, \{Y_{i,1}(A_1)\}_{i \in \mathcal{U}}, \ldots, A_{t-1}, \{Y_{i,t-1}(A_{t-1})\}_{i \in \mathcal{U}}\}$. Specifically, $\pi_t(A) = \mathbb{P}(A_t = A \mid \mathcal{H}_{t-1})$. The performance of the policy is commonly measured by the cumulative regret [Auer et al., 2002a, Lattimore and Szepesvári, 2020], defined as

$$\mathcal{R}(T, \pi) := \max_{A \in \mathcal{K}^{\mathcal{U}}} \sum_{t=1}^{T} \frac{1}{N} \sum_{i \in \mathcal{U}} Y_{i,t}(A) - \mathbb{E}_\pi \left[ \sum_{t=1}^{T} \frac{1}{N} \sum_{i \in \mathcal{U}} Y_{i,t}(A_t) \right].$$

The above-mentioned problem is far more challenging than the simple MAB problem (which only involves $K$ arms), as it involves $K^N$ possible super-arms, increasing the action space exponentially. As shown by Zhang and Wang [2024] (see their Proposition 1), in certain difficult situations, any valid policy $\pi$ will incur regret that grows linearly with the time horizon $T$, i.e., $\mathcal{R}(T, \pi) = \Omega(T)$. To manage this complexity, we adopt the method of Zhang and Wang [2024], employing two key techniques: exposure mapping and clustering [Leung, 2022a, Zhang and Wang, 2024] to reduce the effective dimensionality of the action space. These techniques enable the formulation of a unified framework, `MAB-N`, which captures a broad spectrum of learning settings.

### 3.2 `MAB-N`

**Exposure mapping [Leung, 2022a]**   Exposure mapping is a common tool in causal inference for network interference that reduces the complexity of treatment assignments in networked settings. The core idea of exposure mapping is to compress these high-dimensional features of neighbors and network structure into a smaller set of exposure categories. Instead of labeling each individual as merely "treated" or "untreated," we assign them an exposure level that reflects the degree or type of influence they experience, such as "having two treated neighbors" or "having at least one treated neighbor." The definition of the exposure mapping follows [Leung, 2022a, Zhang and Wang, 2024]:

$$s \equiv \mathbf{S}(i, A, \mathbb{H}), \text{ where } \mathbf{S} : \mathcal{U} \times \mathcal{K}^{\mathcal{U}} \times \{0,1\}^{N \times N} \to \mathcal{U}_s, \tag{1}$$

where $s$ denotes the exposure arm, $\mathbf{S}$ the exposure mapping, and $\mathcal{U}_s$ its output space, referred to as the exposure-arm set, with cardinality $|\mathcal{U}_s| = d_s$. Intuitively, the exposure mapping reduces the original super arm space of size $K^N$ to an exposure arm space of size $d_s$. We define $S = \{\mathbf{S}(i, A, \mathbb{H})\}_{i \in \mathcal{U}} \equiv (s_1, \ldots, s_N)$ as the *exposure super arm*. This allows us to decompose the policy $\pi_t(\cdot)$ and define the expected exposure mapping-based reward:

$$\pi_t(A) \equiv \mathbb{P}(A_t = A \mid S_t)\mathbb{P}(S_t \mid \mathcal{H}_{t-1}), \quad \tilde{Y}_{i,t}(S_t) := \sum_{A \in \mathcal{K}^{\mathcal{U}}} Y_{i,t}(A)\mathbb{P}(A_t = A \mid S_t). \tag{2}$$

Here $S_t$ denotes the exposure super arm selected by the algorithm in round $t$ based on the history $\mathcal{H}_{t-1}$, and the policy $\pi_t(A)$ is represented by a two-stage sampling procedure: it first draws $S_t \sim \mathbb{P}(\cdot \mid \mathcal{H}_{t-1})$ and then samples $A_t \sim \mathbb{P}(\cdot \mid S_t)$. The second line of the above equation is a generalized notation of Leung [2022a]. Notably, $\mathbb{P}(A_t = A \mid S)$ represents a fixed sampling rule that can be manually defined by the learner before the learning starts. Typically, the probability of selecting $A_t = A$ given $S$ is zero if $S$ does not match the set $\{\mathbf{S}(i, A, \mathbb{H})\}_{i \in \mathcal{U}}$. Conversely, if $S$ is equal to this set, then the probability of choosing $A$ is strictly positive, i.e., $\mathbb{P}(A_t = A \mid S) > 0$. In this context, the expected reward of $S$ (i.e., $\tilde{Y}_{i,t}(S)$) depends solely on the definition of the exposure mapping $\mathbf{S}$ and the network topology $\mathbb{H}$.

**Clustering.** We define the clustering set as $\mathcal{C} := \{\mathcal{C}_q\}_{q \in [C]}$, where $C = |\mathcal{C}|$ represents the total number of clusters. The clusters are assumed to be disjoint, meaning that for any $i \neq j, \mathcal{C}_i \cap \mathcal{C}_j = \varnothing$, and collectively exhaustive, such that $\bigcup_{q \in [C]} \mathcal{C}_q = [N]$. For any $i \in [N]$, we denote $\mathcal{C}^{-1}(i)$ as the cluster containing $i$. Such an operation is common and necessary, otherwise, the total arm space is exponentially large.

**Framework of** `MAB-N`. We define the legitimate exposure super arm set as $\mathcal{U}_{\mathcal{E}} := \mathcal{U}_{\mathcal{C}} \cap \mathcal{U}_{\mathcal{O}}$, where $\mathcal{U}_{\mathcal{O}} := \{\{\mathbf{S}(i, A, \mathbb{H})\}_{i \in \mathcal{U}} : A \in \mathcal{K}^{\mathcal{U}}\}$ ensuring that $S \in \mathcal{U}_{\mathcal{O}}$ is compatible with the original arm set $\mathcal{K}^{\mathcal{U}}$, and $\mathcal{U}_{\mathcal{C}} := \{S_t : \forall i, j \in \mathcal{U}, \text{ if } \mathcal{C}^{-1}(i) = \mathcal{C}^{-1}(j), \text{ then } s_{i,t} = s_{j,t}\}$ denoting all kinds of cluster-wise switchback exposure super arms. For instance, if $\mathcal{U}_s \in \{0, 1\}, N = 4, \mathcal{C}_1 = \{1, 2\}, \mathcal{C}_2 = \{3, 4\}$, then $\mathcal{U}_{\mathcal{C}} = \{(k_1, k_1, k_2, k_2) : k_1, k_2 \in \{0, 1\}\}$. Hence, the cardinality of the exposure super arm space satisfies $|\mathcal{U}_{\mathcal{E}}| \leq |d_s|^C$. The word "legitimate" means in each round, the policy can only select an exposure super arm $S_t$ in $\mathcal{U}_{\mathcal{E}}$ and sample the $A_t$ according to $\mathbb{P}(A_t = A \mid S_t)$. The exposure mapping (which controls $d_s$) and clustering (which controls $C$) allow us to manage the action space; they only need to satisfy the following condition:

*Condition* 3.1. The exposure mapping $\mathbf{S}$ and $\mathcal{C}$ should ensure that $2 \leq |\mathcal{U}_{\mathcal{E}}| \leq T$.

In addition, we define $\mathcal{Y}_t(S) = \frac{1}{N} \sum_{i \in \mathcal{U}} \tilde{Y}_{i,t}(S)$ as the expected average reward of the exposure super arm $S \in \mathcal{U}_{\mathcal{E}}$ in round $t$. The reward in round $t$ $R_t(S_t)$ follows $R_t(S_t) = \frac{1}{N} \sum_{i \in \mathcal{U}} Y_{i,t}(A_t)$, where $A_t \sim \mathbb{P}(A_t = A \mid S_t)$.

`MAB-N` **is a unified framework.** It is important to note that `MAB-N` is not parallel to the learning settings in Jia et al. [2024], Agarwal et al. [2024]; rather, it provides a more general framework that encompasses these settings. In the following, we provide several illustrative examples: **Example (i).** *Classic MAB* [Auer et al., 2002b, Simchi-Levi and Wang, 2024] corresponds to the case where $N = 1$, that is, a single unit without network effects, and the exposure mapping is defined as $\mathbf{S}(1, A, \mathbb{H}) := A$, where $A \in \mathcal{K}$. **Example (ii).** Agarwal et al. [2024] adopt an exposure mapping of the form $\mathbf{S}(i, A, \mathbb{H}) := Ae_i$ and set $C = N$, meaning each unit is assigned to its own cluster. **Example (iii).** Jia et al. [2024] also define $\mathbf{S}(i, A, \mathbb{H}) := Ae_i$, but use $C = 1$, assigning all units to a single cluster. This models the global proportion of treatment at each round $t$. **Example (iv).** The exposure mapping and clustering framework can also be traced back to the offline causal inference literature. Suppose $\sum_j h_{ij} > 0$ for all $j \in \mathcal{U}$. The exposure mapping can be defined as $\mathbf{S}(i, A, \mathbb{H}) := \mathbf{1} \left\{ \frac{\sum_{j \in \mathcal{U}} h_{ij} a_j}{\sum_{j \in \mathcal{U}} h_{ij}} \in \left[0, \frac{1}{2}\right) \right\}$, which is adapted from the offline setting [Leung, 2022a, Gao and Ding, 2023].

As shown in the above examples, `MAB-N` is a unified framework that captures a wide range of learning settings. Studying it effectively subsumes many existing scenarios. For example, to model a switchback design as in Jia et al. [2024], one can adopt the exposure mapping and clustering described in Example (iii). Moreover, `MAB-N` also enables the exploration of novel frameworks that have not been considered in prior online settings, such as the one presented in Example (iv).

**Is $\mathbb{H}$ necessarily known?** We emphasize that whether the adjacency matrix $\mathbb{H}$ must be known a priori depends entirely on how the exposure mapping $\mathbf{S}(\cdot)$ is defined. For example, if our setting reduces to the scenario in Leung [2022a]—namely, when the exposure mapping depends on all first-order neighbours—then the neighbourhood information in $\mathbb{H}$ must indeed be known in advance. In contrast, if our exposure mapping simply uses each node's cluster index, then $\mathbb{H}$ can remain unknown. Overall, we include $\mathbb{H}$ in our setup in order to focus on a unified framework, and this does not imply that all information in $\mathbb{H}$ always needs to be learned.

Finally, we introduce the definition of the ATE [Leung, 2022a, Liang and Bojinov, 2023].

**Definition 3.2** (ATE). The ATE between exposure super arm $S_i, S_j \in \mathcal{U}_{\mathcal{E}}$ in round $t$ is $\bar{\tau}_t(S_i, S_j) = \frac{1}{t}\sum_{t'=1}^{t} \tau_t(S_i, S_j) = \frac{1}{t}\sum_{t'=1}^{t} \left(\mathcal{Y}_{t'}(S_i) - \mathcal{Y}_{t'}(S_j)\right)$.

### 3.3 Learning objectives

**Objective 1: Regret minimization.** Based on the setting of the `MAB-N`, we can refine the definition of regret in Section 3.1:

$$\mathcal{R}(T, \pi) = \max_{S \in \mathcal{U}_{\mathcal{E}}} \sum_{t=1}^{T} \mathcal{Y}_t(S) - \mathbb{E}_{\pi}\left[\sum_{t=1}^{T} R_t(S_t)\right].$$

**Objective 2: Continual inference.** For all $S_i, S_j \in \mathcal{U}_{\mathcal{E}}$, our objective is to design a $(1 - \tilde{\delta})$ CS $\{I_t(S_i, S_j)\}_{t=1}^{\infty}$, where each $I_t(S_i, S_j)$ is an interval and $\tilde{\delta}$ is the probability parameter such that $\mathbb{P}\left(\forall t \geq 1, \bar{\tau}_t(S_i, S_j) \in I_t(S_i, S_j)\right) \geq 1 - \tilde{\delta}$.

**Objective 3: ATE estimation error minimization.** We aim to design estimators $\hat{\Delta}_T(S_i, S_j)$ for all $S_i, S_j \in \mathcal{U}_{\mathcal{E}}$, to minimize the maximum ATE estimation error [Simchi-Levi and Wang, 2024, Zhang and Wang, 2024] defined as $e_{\nu}(T, \hat{\Delta}) = \max_{S_i, S_j \in \mathcal{U}_{\mathcal{E}}} \mathbb{E}\left[|\hat{\Delta}_T(S_i, S_j) - \bar{\tau}_T(S_i, S_j)|\right]$.

Recalling Figure 1, simultaneously addressing Obj. 1-3 is a shared concern among online learning and statistical researchers, with the former primarily focusing on Obj. 1 and the latter on Obj. 2-3. However, achieving both Obj. 1 and Obj. 2-3 simultaneously is often challenging; essentially, there is a trade-off between the two. When we excessively prioritize the estimator's accuracy (e.g., through independent random sampling), we may fail to adequately explore the optimal strategy, resulting in regret that it does not remain sublinear in time $T$. Conversely, if we focus solely on identifying the optimal strategy, we naturally overlook the measurement of the reward gap, which can lead to uncontrolled variance in the estimator. In the following section, we provide a rigorous description of the relationship between Objective 1 and Objective 3.

## 4 Pareto optimality results

In this paragraph, we aim to construct the theoretical optimal trade-off between the ATE estimation accuracy and the regret.

**Theorem 4.1.** *Given any online decision-making policy $\pi$, and any $\mathcal{S}$ and $\mathcal{C}$ that satisfy Condition 3.1, the trade-off between regret and ATE estimation exhibits*

$$\inf_{\hat{\Delta}} \max_{\nu \in \mathcal{E}_0} \left(\sqrt{\mathcal{R}_{\nu}(T, \pi)} e_{\nu}(T, \hat{\Delta})\right) = \Omega_{K,T}(\sqrt{|\mathcal{U}_{\mathcal{E}}|}), \tag{3}$$

*where $\mathcal{R}_{\nu}$ and $e_{\nu}$ denote, respectively, the regret and the maximum ATE estimation error under instance $\nu$.*

**The sketch of proof.** To prove the lower bound, we construct two adversarial bandit instances that differ only in the expected reward associated with one exposure super arm $S$, while all other distributions remain identical. This creates a small but fixed difference in the average treatment effect (ATE) between $S$ and another arm $S'$, yet renders the two instances statistically hard to distinguish. The crux of the argument is that unless the learner samples $S$ sufficiently often, it cannot accumulate enough information to detect this perturbation. Consequently, any estimator of the ATE between $S$ and $S'$ will exhibit a large error due to insufficient exploration. Formally, the argument applies tools from information theory, specifically Le Cam's two-point method and a KL-divergence bound, to show that accurate estimation of the ATE requires distinguishing between the two constructed environments, which in turn necessitates a minimum number of pulls of arm $S$. This induces a direct tension between the estimation error $e(T, \hat{\Delta})$ and the cumulative regret $\mathcal{R}(T, \pi)$: minimizing one forces the other to grow. By optimizing this trade-off, we derive the lower bound as above, which highlights a fundamental information-theoretic limit in adversarial bandits under network interference. The novelty lies in extending classical bandit lower-bound techniques to the setting of

networked exposure mappings and adversarial reward generation, preserving the sharp dependency on the effective arm space size $|\mathcal{U}_{\mathcal{E}}|$.

Theorem 4.1 establishes the fundamental trade-off between the estimation, namely, the statistical power, and the cumulative regret, namely, the learning efficiency. For instance, when the estimation achieves $T^{-1/2}$ estimation, we claim that, unfortunately, the regret will exhibit as $\Omega(T)$. In contrast, when we omit the estimation of ATE and solely figure out the best arm, the regret will converge. This guideline essentially encourages practitioners to carefully and reasonably design estimators and evaluate their convergence performance concerning $T$. When practitioners are more inclined to estimate the reward gap between different arms rather than pursuing the optimal policy — such as in scenarios where hospitals, during a specific period of a pandemic, aim to assess the efficacy of treatments more accurately — efforts should be directed toward actively designing estimators with higher convergence efficiency. Practitioners should also be prepared to accept the trade-off of potential losses in regret convergence resulting from this approach.

# 5 Asymptotic Confidence Sequence and Main Algorithm

In this section, we first introduce a technique called asymptotic CS, which facilitates continual inference of the ATE as defined in Definition 3.2. Next, we propose our algorithm EXP3-N-CS that integrates asymptotic CS to achieve three objectives.

## 5.1 Asymptotic CS and MAD

CS is a series of confidence intervals that remain uniformly valid over time [Darling and Robbins, 1967, Waudby-Smith et al., 2021]. Unlike traditional confidence intervals, which are limited to inference at a pre-specified terminal time $T$, a CS enables continual inference throughout the process. This allows for adaptive decisions regarding experiment termination or continuation, as the learning algorithm does not need to know or define the time horizon $T$ in advance. Instead, the algorithm can continuously utilize the CS for inference, concluding the experiment once satisfactory learning outcomes are achieved. We introduce the concept of asymptotic CS, first developed by Waudby-Smith et al. [2021].

**Definition 5.1** (Asymptotic $(1 - \tilde{\delta})$ CS). Suppose there exists an (unknown) non-asymptotic $(1 - \tilde{\delta})$ CS $\{\hat{\mu}_t \pm C_t^*\}_{t=1}^{\infty}$ for a sequence of target parameter $\{\mu_t\}_{t=1}^{\infty}$ and a CS $\{\hat{\mu}_t \pm \hat{C}_t\}_{t=1}^{\infty}$ such that $\frac{\hat{C}_t}{C_t^*} \xrightarrow{a.s.} 1$, then $\{\hat{\mu}_t \pm \hat{C}_t\}_{t=1}^{\infty}$ is an asymptotic $(1 - \tilde{\delta})$ CS for $\{\mu_t\}_{t=1}^{\infty}$.

Our Asymptotic CS for MAB-N is defined in the following proposition.

**Proposition 5.2** (Asymptotic CS for MAB-N). *We define the asymptotic CS for $S_i, S_j \in \mathcal{U}_{\mathcal{E}}$ as $\{\hat{\bar{\tau}}_t(S_i, S_j) \pm \hat{C}_t(S_i, S_j)\}_{t=1}^{\infty}$. The IPW estimator $\hat{\bar{\tau}}_t(S_i, S_j)$ is defined as $\frac{1}{t}\sum_{t'=1}^{t} \hat{\tau}_{t'}(S_i, S_j) = \frac{1}{t}\sum_{t'=1}^{t}\left(\frac{\mathbf{1}\{S_{t'}=S_i\}R_{t'}(S_{t'})}{\pi_{t'}(S_i)} - \frac{\mathbf{1}\{S_{t'}=S_j\}R_{t'}(S_{t'})}{\pi_{t'}(S_j)}\right)$, which serves to estimate $\bar{\tau}_t(S_i, S_j)$. The CS width $\hat{C}_t(S_i, S_j) = \sqrt{\frac{2(\hat{\mathcal{V}}_t(S_i,S_j)\eta^2+1)}{t^2\eta^2}\log\left(\frac{\sqrt{\hat{\mathcal{V}}_t(S_i,S_j)\eta^2+1}}{\tilde{\delta}}\right)}$, where $\hat{\mathcal{V}}_t(S_i, S_j) = \sum_{t'=1}^{t}\left(\frac{1}{\pi_{t'}^{MAD}(S_i)} + \frac{1}{\pi_{t'}^{MAD}(S_j)}\right)$ and $\eta$ is an arbitrary positive parameter.*

Asymptotic CS can appear as a plug-in module that operates independently of any specific algorithm. Its performance is based on the following assumption:

**Assumption 5.3.** We require that the cumulative conditional variances $\mathcal{V}_t(S_i, S_j) = \sum_{t'=1}^{t} \mathbf{V}(\hat{\tau}_{t'}(S_i, S_j) \mid \mathcal{F}_{t'})$ grow at least linearly with $t$ for all $S_i, S_j \in \mathcal{U}_{\mathcal{E}}$, that is, $\mathcal{V}_t(S_i, S_j) = \Omega(t)$, where $\mathcal{F}_t$ denotes the sigma algebra that contains $\{\tilde{Y}_{i,t'}(S)\}_{S\in\mathcal{U}_{\mathcal{E}}, i\in\mathcal{U}, t'\in[t]}$ and $\mathcal{H}_t$.

The above assumption is weaker than the one made by Simchi-Levi and Wang [2024], which assumes that the expected reward gap of each pair of arms is $\Theta(1)$ (stochastic setting). We should mention that our assumption is relatively stronger than the assumption in Waudby-Smith et al. [2021], Ham et al. [2023], which only requires the cumulative conditional variance $\mathcal{V}_t(S_i, S_j) \to \infty$ when $t \to \infty$, but does not assume a linear growth rate. However, this assumption should hold in the most realistic experimental settings, provided that instances where there exists a time $t'$ such that $\exists S, \mathcal{Y}_t(S) = 0$

---

**Algorithm 1** EXP3-N-CS

---

1: **Input:** arm set $\mathcal{A}$, unit set $\mathcal{U}$, exposure super arm set $\mathcal{U}_\mathcal{E}$, sequence $\{\mathcal{L}_m\}_{m=1}^\infty$
2: **for** $t = 1, 2, \ldots$ **do**
3:  Compute $m$ such that $t \in \mathcal{L}_m$, set $\epsilon_m = \sqrt{\frac{\log(|\mathcal{U}_\mathcal{E}|)}{|\mathcal{U}_\mathcal{E}|2^{m-1}}}$
4:  **if** $t = t_m$ **then**
5:   For all $S \in \mathcal{U}_\mathcal{E}$, set $\pi_t^{\text{ALG}}(S) = \frac{1}{|\mathcal{U}_\mathcal{E}|}$
6:  **else**
7:   For all $S \in \mathcal{U}_\mathcal{E}$, set $\pi_t^{\text{ALG}}(S) = \frac{\exp\left(\epsilon_m \hat{R}_{\mathcal{L}_m, t-1}(S)\right)}{\sum_{S' \in \mathcal{U}_\mathcal{E}} \exp\left(\epsilon_m \hat{R}_{\mathcal{L}_m, t-1}(S')\right)}$
8:  **end if**
9:  For all $S \in \mathcal{U}_\mathcal{E}$, set $\pi_t^{\text{MAD}}(S) = \frac{1}{|\mathcal{U}_\mathcal{E}|}\delta_t + (1 - \delta_t)\pi_t^{\text{ALG}}(S)$
10:  Sample $S_t$ based on $\pi_t^{\text{MAD}}$, implement $\texttt{Sampling}(S_t)$ and observe the rewards $R_t(S_t)$
11:  For all $S_i, S_j \in \mathcal{U}_\mathcal{E}$, construct the confidence sequence $\hat{\bar{\tau}}_t(S_i, S_j) \pm \hat{C}_t(S_i, S_j)$
12:  For all $S \in \mathcal{U}_\mathcal{E}$, set $\hat{R}_{\mathcal{L}_m, t}(S) = \sum_{t'=t_m}^t 1 - \frac{\mathbf{1}\{S_t = S\}(1 - R_t(S_t))}{\pi_t^{\text{MAD}}(S)}$
13: **end for**
14: Return $\hat{\Delta}_T^{(i,j)} = \hat{\bar{\tau}}_T(S_i, S_j)$ for all $S_i, S_j \in \mathcal{U}_\mathcal{E}$

---

for all $t > t'$ are rare in practice and may indicate practical problems with the experiment [Liang and Bojinov, 2023]. The asymptotic CS presented in Proposition 5.2 can be incorporated into various classic adversarial bandit algorithms, such as EXP3. However, algorithms like EXP3 primarily focus on minimizing regret, which often leads to sampling the exposure super arm with low rewards at a low probability. This behavior can reduce the accuracy of our IPW estimator in estimating low-reward exposure super arms and significantly weaken the inference power of the asymptotic CS. Therefore, it is essential to ensure that the algorithm explores each exposure super arm with sufficiently high probability. To this end, we incorporate the MAD [Liang and Bojinov, 2023], a modular component that can be integrated into various algorithms to promote effective exploration.

**Definition 5.4** (MAD). Let the probability of Algorithm ALG pulling arm $S$ in round $t$ be denoted by $\pi_t^{\text{ALG}}(S) = \mathbb{P}_{\text{ALG}}(S_t = S \mid \mathcal{H}_{t-1})$, where $\mathbb{P}_{\text{ALG}}$ denotes the probability taken with respect to ALG. After applying MAD, the probability of pulling the exposure super arm $S$ in round $t$ is given by $\pi_t^{\text{MAD}}(S) = \mathbb{P}_{\text{MAD}}(S_t = S \mid \mathcal{H}_{t-1}) = \frac{1}{|\mathcal{U}_\mathcal{E}|}\delta_t + (1 - \delta_t)\pi_t^{\text{ALG}}(S)$, where $\delta_t \in [0, 1]$ is a time-varying parameter and $\mathbb{P}_{\text{MAD}}$ denotes the probability taken concerning MAD. It is easy to verify that $\pi_t^{\text{MAD}}(S) \in [0, 1]$ for all $S \in \mathcal{U}_\mathcal{E}$ and $\sum_{S \in \mathcal{U}_\mathcal{E}} \pi_t^{\text{MAD}}(S) = 1$.

MAD can balance the trade-off between regret minimization and additional exploration. Consider two special cases: $\delta_t = 0$ and $\delta_t = 1$. When $\delta_t = 0$, the policy becomes $\pi_t^{\text{MAD}} = \pi_t^{\text{ALG}}$, entirely focusing on minimizing regret (as we suppose ALG intends to minimize the regret). On the other hand, when $\delta_t = 1$, the policy becomes $\pi_t^{\text{MAD}}(S) = \frac{1}{|\mathcal{U}_\mathcal{E}|}$ for all $S \in \mathcal{U}_\mathcal{E}$ (uniformly samples $S_t$ from $\mathcal{U}_\mathcal{E}$), entirely prioritizing exploration. The following Theorem 5.5 shows that with a specific setup, the CS proposed in Proposition 5.2 is a valid asymptotic $(1 - \tilde{\delta})$ CS.

**Theorem 5.5** (Performance of the Asymptotic CS). *Suppose* $\mathbf{S}$ *and* $\mathcal{C}$ *satisfy Condition 3.1 and Assumption 5.3 holds. For all* $S_i, S_j \in \mathcal{U}_\mathcal{E}$, *consider the sequence of random variables* $\left(\hat{\tau}_t(S_i, S_j)\right)_{t=1}^\infty$, *where the probability of observing* $S_t = S$ *at time* $t$ *is given by* $\pi_t^{MAD}(S) = \frac{1}{|\mathcal{U}_\mathcal{E}|}\delta_t + (1 - \delta_t)\pi_t^{ALG}(S)$. *We set* $\delta_t = \frac{1}{t^\alpha}$ *which satisfies* $\alpha \in [0, \frac{1}{2})$. *The CS in Proposition 5.2 forms a valid asymptotic CS for* $(\bar{\tau}_t(S_i, S_j))_{t=1}^\infty$ *with confidence level* $1 - \tilde{\delta}$ *and the CS width* $\hat{C}_t(S_i, S_j) = \tilde{O}(|\mathcal{U}_\mathcal{E}|^{\frac{1}{2}} t^{\frac{\alpha-1}{2}})$.

### 5.2 Main algorithm

In the previous section, we demonstrated that both the asymptotic CS and the MAD can be integrated into learning algorithms such as EXP3. In this section, we analyze the performance of the resulting algorithm, which we refer to as EXP3-N-CS.

EXP3-N-CS is designed to achieve three learning objectives. We ensure that the algorithm does not rely on prior knowledge of $T$ by employing the doubling trick [Besson and Kaufmann, 2018a]. We define the time interval $\mathcal{L}_m := \{t_m, \ldots, t_m + 2^{m-1} - 1\}$, where $t_1 = 1$ and $t_m = 1 + \sum_{m'=0}^{m-2} 2^{m'}$

---

**Algorithm 2** `Sampling`

---

1: **Input:** $S_t$
2: Derive the set of real super arm $\{Z_{l'}\}_{l' \in [l]}$ such that for all $Z_{l'}$, $\{\mathbf{S}(i, Z_{l'}, \mathcal{H})\}_{i \in \mathcal{U}} = S_t$
3: Sample $A_t$ from set $\{Z_{l'}\}_{l' \in [l]}$ based on $\mathbb{P}(A_t = Z_{l'} \mid S_t)$, pull $A_t$, and observe reward $R_t(S_t) = \frac{1}{N} \sum_{i \in \mathcal{U}} Y_{i,t}(A_t)$

---

for all $m > 1$. The algorithm begins by computing the policy $\pi_t^{\text{ALG}}(S)$ (ALG equals to EXP3) for all $S \in \mathcal{U}_\mathcal{E}$ using the standard EXP3 technique (line 4-8). Next, it adjusts the policy to derive $\pi_t^{\text{MAD}}(S)$ for all $S \in \mathcal{U}_\mathcal{E}$ based on the MAD (line 9). The algorithm then samples $S_t$ based on $\pi_t^{\text{MAD}}$ and subsequently samples $A_t$ conditioned on $S_t$ (line 10 and Algorithm 2). Algorithm 2 will first derive a real super arm candidate set $\{Z_{l'}\}_{l' \in [l]}$ such that $\{\mathbf{S}(i, Z_{l'}, \mathcal{H})\}_{i \in \mathcal{U}} = S$, $\forall l' \in [l]$. Then, it will sample $A_t$ from $\{Z_{l'}\}_{l' \in [l]}$ based on $\mathbb{P}(A_t = A \mid S_t)$ (note that for all $A \notin \{Z_{l'}\}_{l' \in [l]}$, $\mathbb{P}(A_t = A \mid S_t) = 0$). Using the asymptotic CS proposed in Proposition 5.2, the algorithm constructs the CS to estimate the ATE in each round (line 11). Note that the $\pi_t(S)$ in the asymptotic CS should be replaced with $\pi_t^{\text{MAD}}(S)$. Finally, after the algorithm terminates the iteration, the algorithm outputs $\hat{\Delta}_T(S_i, S_j) = \hat{\bar{\tau}}_T(S_i, S_j)$ as the estimated ATE (line 14).

**Theorem 5.6.** *Following the setting in Theorem 5.5:*

*(i) (**Estimation error upper bound**) For all $S_i, S_j \in \mathcal{U}_\mathcal{E}$, define $\hat{\Delta}_T(S_i, S_j) := \hat{\bar{\tau}}_T(S_i, S_j)$. Then, we have $\mathbb{E}[|\hat{\Delta}_T(S_i, S_j) - \bar{\tau}_T(S_i, S_j)|] = \tilde{O}\big(|\mathcal{U}_\mathcal{E}|^{\frac{1}{2}} t^{\frac{\alpha-1}{2}}\big)$.*

*(ii) (**Regret upper bound**) The regret of the `EXP3-N-CS` can be upper bounded by $\mathcal{R}(T, \pi^{MAD}) = \tilde{O}\big(\sqrt{|\mathcal{U}_\mathcal{E}|T} + T^{1-\alpha}\big)$.*

*(iii) (**Pareto-optimality**) For all legitimate instances $\nu \in \mathcal{E}_0$, select $\alpha$ such that $\sqrt{|\mathcal{U}_\mathcal{E}|T} \leq T^{1-\alpha}$ and $\alpha \in [0, \frac{1}{2})$, then `EXP3-N-CS` guarantees $e_\nu(T, \hat{\Delta}) \sqrt{\mathcal{R}_\nu(T, \pi^{MAD})} = \tilde{O}\big(\sqrt{|\mathcal{U}_\mathcal{E}|}\big)$.*

From Theorem 5.6 (iii), we conclude that `EXP3-N-CS` achieves the Pareto-optimal trade-off established in Theorem 4.1. There is no need to choose $\alpha$ larger than $\frac{1}{2}$, as doing so does not reduce the regret but instead deteriorates the estimation accuracy. Furthermore, although our analysis centers on `EXP3-N-CS`, the underlying design principles naturally extend to a broader class of bandit algorithms, owing to the strong modularity and composability of the Asymptotic CS and MAD components. In particular, for any base algorithm that achieves a regret bound of the form $\tilde{O}(\sqrt{|\mathcal{U}_\mathcal{E}|T})$ under the `MAB-N` framework, the performance guarantee in Theorem 5.6 can be easily extended. This is because the MAD adopts the form $\pi_t^{\text{MAD}}(S) = \frac{1}{|\mathcal{U}_\mathcal{E}|}\delta_t + (1-\delta_t)\pi_t^{\text{ALG}}(S)$, where the first term (with coefficient $\delta_t$) is introduced to improve ATE estimation, and the second term $\pi_t^{\text{ALG}}(S)$ is the base algorithm aimed at regret minimization. To analyze the regret, we decompose the total regret into two components corresponding to the two terms in the strategy, and then separately upper bound each. As shown in Theorem 5.6 (ii), the first part of the regret scales as $\tilde{\mathcal{O}}(\sqrt{|\mathcal{U}_\mathcal{E}|T})$, which comes from the $(1-\delta_t)\pi_t^{\text{ALG}}(S)$ term, and the second part scales as $\tilde{\mathcal{O}}(T^{1-\alpha})$, arising from the $\frac{1}{|\mathcal{U}_\mathcal{E}|}\delta_t$ term, which is independent of the specific base algorithm. Therefore, we can easily analyze the overall regret upper bound as $\tilde{\mathcal{O}}\big(\sqrt{|\mathcal{U}_\mathcal{E}|T} + T^{1-\alpha}\big)$. The Asymptotic CS and MAD components serve as general-purpose mechanisms that facilitate balancing Objectives 1–3 across a broad range of algorithms.

We now present guidance on selecting $\alpha$ by combining theoretical insights with practical considerations. Specifically, setting $\alpha = 0$ results in linear regret, but ensures a fast convergence rate for the CS width and the ATE, specifically $\tilde{O}\big(|\mathcal{U}_\mathcal{E}|^{\frac{1}{2}} t^{-\frac{1}{2}}\big)$. In addition, setting $\alpha$ such that $\sqrt{|\mathcal{U}_\mathcal{E}|T} = T^{1-\alpha}$, we can minimize the regret to the level of $\tilde{O}\big(\sqrt{|\mathcal{U}_\mathcal{E}|T}\big)$ while achieving statistical inference with $\tilde{O}\big(|\mathcal{U}_\mathcal{E}|^{\frac{1}{4}} T^{-\frac{1}{4}}\big)$. To demonstrate its practical selection, consider the following example. When treating a group of critically ill patients, the primary goal is to minimize regret by assigning treatments that are currently believed to be most effective, thereby improving their immediate survival chances within the network. Conversely, for patients with milder symptoms and stable conditions, it can be advantageous to explore less-certain yet promising treatments, as doing so improves the evaluation of the relative effectiveness of different options. This enhanced inference leads to more precise ATE estimation and, in turn, supports better-informed treatment decisions for future populations. Accordingly, one may

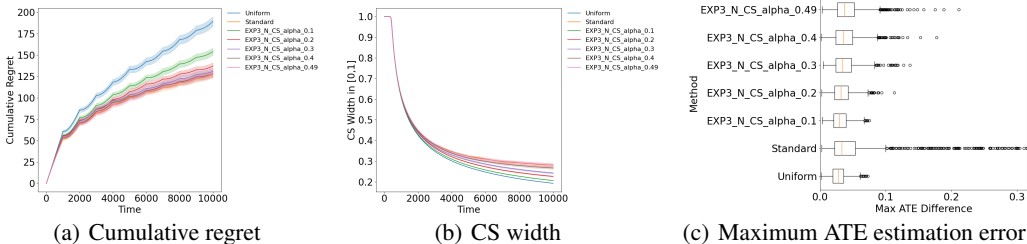

| (a) Cumulative regret | (b) CS width | (c) Maximum ATE estimation error |

Figure 2: Experimental results.

prefer selecting $\alpha$ closer to $\frac{1}{2}$ in the former scenario, prioritizing regret minimization, and closer to $0$ in the latter, emphasizing pure exploration.

## 6  Experiments

In this section, we demonstrate the empirical performance of our `EXP3-N-CS` by some simulation studies. The code is available at: `https://github.com/TheoryMagic/Design-based-Bandits`.
**Setup.** We consider a network consisting of 101 units. Specifically, there is one center cluster $C_1 = \{1\}$ that contains a single unit, which is connected to every unit in the five outer clusters. Each of the outer clusters contains 20 units. We set the action set $\mathcal{K} = \{0, 1\}$. Additionally, we define the exposure mapping inspired by [Leung, 2022a, Gao and Ding, 2023], expressed as $\mathbf{S}(i, A, \mathbb{H}) = \mathbf{1}\left\{\frac{\sum_j h_{i,j} \times a_j}{\sum_j h_{i,j}} \in \left[0, \frac{1}{2}\right)\right\}$, exploring the influence of the proportion of action 1 taken among all the neighbors of each unit. The exposure mapping implies $d_s = 2$. For all $S \in \mathcal{U}_\mathcal{E}$, we define $\mathbb{P}(A_t = A \mid S)$ as uniform sampling, and $\mathcal{Y}_t(S) = \frac{1}{N}\sum_{i \in \mathcal{U}} Y_{i,t}(A)$ for all $A$ such that $\{\mathbf{S}(i, A, \mathbb{H})\}_{i \in \mathcal{U}} = S$. Besides, we let $\mathcal{Y}_t(S)$ be sampled from a Bernoulli distribution. The mean of this Bernoulli distribution is uniformly resampled from $[0, 1]$ every 1000 rounds. We set the trade-off parameter of `EXP3-N-CS` to $\alpha \in \{0.1, 0.2, 0.3, 0.4, 0.49\}$ and compare its performance against two baselines: *Standard* (where $\delta_t = 0$) and *Uniform* (where $\delta_t = 1$). Each algorithm is executed 1000 times, and we report the averaged results. **Results.** The simulation results are shown in Fig. 2(a), 2(b) and 2(c). From Fig. 2(a), the Uniform baseline consistently exhibits the highest cumulative regret throughout the entire horizon. In contrast, both the Standard baseline and `EXP3-N-CS` with larger $\alpha$ values (e.g., $\alpha = 0.4$ or $\alpha = 0.49$) achieve the lowest cumulative regret. This is because Uniform does not focus on minimizing regret. Fig. 2(b) illustrates the trajectories of the CS width $\hat{C}_t(S_i, S_j)$, where $S_i, S_j = \arg\max_{S_i, S_j \in \mathcal{U}_\mathcal{E}} \hat{C}_T(S_i, S_j)$ ($\hat{C}_T(S_i, S_j)$ takes the average value of 1000 times). The Uniform baseline achieves the narrowest CS, indicating the most accurate inference. In contrast, the Standard baseline maintains the widest CS width throughout the horizon, implying the least accurate inference. The `EXP3-N-CS` variants lie between these two extremes, with smaller $\alpha$ values producing wider CS widths that approach that of Uniform. Fig. 2(c) presents the box plot of the maximum ATE estimation error (i.e., $e_\nu(T, \hat{\Delta})$), where the orange line represents the median. As shown in Fig. 2(c), both `EXP3-N-CS` variants with smaller $\alpha$ values and the Uniform baseline achieve relatively low maximum ATE estimation errors with compact interquartile ranges and fewer extreme outliers. In contrast, the Standard baseline exhibits a noticeably wider spread of errors and a substantial number of outliers. This inferior inference performance of Standard (Obj. 2–3) is attributed to its lower frequency of exploring sub-optimal arms compared to *Uniform* and the `EXP3-N-CS` variants. Due to page limitations, we present four extensive experimental instances in Section F of the Appendix.

## Acknowledgements

Haoyang Hong and Huazheng Wang are supported by National Science Foundation under grant IIS-2403401. Zhiheng Zhang is supported by "the Fundamental Research Funds for the Central Universities" (number: 2025110602) of Shanghai University of Finance and Economics.

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

# A   Notations

| | |
|---|---|
| $\mathcal{U}$ | Set of units |
| $N$ | Number of units |
| $\mathbb{H}$ | Adjacency matrix |
| $\mathcal{K}$ | Real arm set (action set) |
| $K$ | Number of real arms |
| $a_{i,t}$ | Arm assigned to unit $i$ |
| $A_t$ | Real super arm pulled in round $t$ |
| $\mathbf{S}(i, A, \mathbb{H})$ | Exposure mapping |
| $s_{i,t}$ | Exposure arm assigned to unit $i$ |
| $S_t$ | Exposure super arm sampled in round $t$ |
| $\hat{R}_{\mathcal{L}_m,t}(S)$ | Reward estimator for exposure super arm $S$ |
| $\mathcal{U}_s$ | Set of exposure super-arms |
| $d_s$ | Number of exposure arm |
| $\mathcal{U}_{\mathcal{E}}$ | Legitimate exposure super arm set |
| $\mathcal{U}_{\mathcal{O}}$ | Set of exposure super arm that can be triggered by real super arm |
| $\mathcal{U}_{\mathcal{C}}$ | Set of cluster-wise switchback exposure super arm |
| $\pi_t^{\mathrm{ALG}}(S)$ | Probability of Algorithm $\mathtt{ALG}$ pulling exposure super arm $S$ |
| $\pi_t^{\mathrm{MAD}}(S)$ | Probability of pulling exposure super arm $S$ after using MAD |
| $\mathcal{E}_0$ | Set of legitimate instances |
| $Y_{i,t}(A)$ | Expected reward of the unit $i$ under $A$ |
| $\tilde{Y}_{i,t}(S)$ | Expected reward of unit $i$ under $S$ |
| $\mathcal{Y}_t(S)$ | Average expected reward under $S$ |
| $R_t(S_t)$ | Average reward under $S_t$ in round $t$ |
| $\mathcal{R}(T, \pi)$ | Cumulative regret |
| $\tau_t(S_i, S_j)$ | Difference between potential outcome of $S_i$ and $S_j$ |
| $\bar{\tau}_t(S_i, S_j)$ | ATE between $S_i$ and $S_j$ |
| $\hat{\tau}_t(S_i, S_j)$ | IPW estimator for $\tau_t(S_i, S_j)$ |
| $\hat{\bar{\tau}}_t(S_i, S_j)$ | IPW estimator for $\bar{\tau}_t(S_i, S_j)$ |
| $\hat{C}_t(S_i, S_j)$ | CS width |
| $\mathcal{V}_t(S_i, S_j)$ | Cumulative conditional variance between $S_i$ and $S_j$ |
| $\hat{\mathcal{V}}_t(S_i, S_j)$ | Estimator of the cumulative conditional variance between $S_i$ and $S_j$ |
| $\{\bar{\tau}_t(S_i, S_j) \pm \hat{C}_t(S_i, S_j)\}_{t=1}^{\infty}$ | Confidence sequence |
| $\hat{\Delta}_T(S_i, S_j)$ | Estimated ATE between $S_i$ and $S_j$ |
| $e_\nu(T, \hat{\Delta})$ | Maximum estimation error of the ATE |

## B Comparing MABNI with Causality, Multiple-Play, Multi-Agent and Combinatorial Bandits

Multi-armed bandits with network interference (MABNI) focus on online decision-making under networked dependencies, where each action on one node affects others' rewards through interference. Our work formalizes this challenge by establishing a theoretical trade-off between regret minimization and inference precision, and by constructing design-based anytime-valid confidence sequences that remain valid under adversarial network structures [Zhang and Wang, 2024]. In contrast, causal inference aims to characterize the identifiability of treatment effects under explicit structural assumptions. When point identification fails, partial identification methods quantify what range of causal effects remain compatible with observed data and plausible assumptions [Zhang and Su, 2024, Zhang, 2024]. This distinction suggests that MABNI primarily concerns *learnability under adaptive design*, whereas causal inference addresses *identifiability under fixed assumptions*. Yet the two are deeply connected: sequential designs that minimize regret can also be optimized to tighten identification regions as data accumulate, thereby transforming the exploration process into a controlled reduction of causal uncertainty. Moreover, robust and proxy-based identification methods [Zhang et al., 2023] complement design-based randomization by mitigating latent confounding in observational segments that interleave with experimental interventions. In settings with interference and dynamic dependencies, such as social or temporal networks, causal structures evolve over time; this connects MABNI to dynamic Granger-style causal discovery [Zhang et al., 2020b]. Finally, the trade-off between exploration, inference, and structure generalizes beyond networks to broader combinatorial domains, where the geometry of feasible allocations—such as simplicial partitions or influence subgraphs—determines the attainable identification set [Su et al., 2023, Zhang, 2022, Zhang et al., 2025]. Taken together, MABNI operationalizes the causal principle of controlled experimentation under interference, while partial and robust causal inference extend its theoretical foundation toward identifiability, providing a unified lens on how to jointly optimize regret, inference validity, and causal learnability within complex, networked environments.

The MABNI problem is related to the *multi-agent bandit* problem, in which multiple agents simultaneously pull arms in each round. These agents often collaborate by sharing their local observations to collectively accelerate learning [Szörényi et al., 2013, Wu et al., 2016, He et al., 2022, Wang et al., 2019, 2023a]. A key distinction lies in the modeling assumptions: multi-agent bandit formulations typically assume a priori relationships among agents—such as cooperation or competition—and place significant emphasis on the design of communication protocols to enable coordination or negotiation. Besides, the *multi-play bandit* problem, where the algorithm selects multiple arms in each round and receives individual reward feedback for each, is closely related to the MABNI setting. This framework has been extensively studied in the literature [Louëdec et al., 2015, Lagrée et al., 2016, Zhou and Tomlin, 2018, Besson and Kaufmann, 2018b, Jia et al., 2023]. While both settings involve the simultaneous selection of multiple actions, MABNI further emphasizes the interference among actions selected at different units, where the reward of a unit may depend not only on its own action but also on the other actions selected in the same round. Furthermore, our work is also related to the *combinatorial bandit* problem, where the learner selects a subset of base arms—often subject to combinatorial constraints such as budgets or matroids—and receives feedback and rewards that depend on the selected combination [Cesa-Bianchi and Lugosi, 2012, Chen et al., 2013, 2014, Combes et al., 2015, Kveton et al., 2015, Saha and Gopalan, 2019, Wang et al., 2023b]. Some existing works consider interference effects among units, but such interference is typically either explicitly known to the learner or assumed to follow a predefined structural pattern. In contrast, the MABNI makes no assumptions about the nature or structure of interference across units; instead, it needs to implicitly learn the interference effects through observed rewards.

## C Proof of Theorem 4.1

*Proof of Theorem 4.1.* Recall that the definition of ATE in round $t$ is defined as

$$\bar{\tau}_t(S_i, S_j) = \frac{1}{t} \sum_{t'=1}^{t} \tau_t(S_i, S_j) = \frac{1}{t} \sum_{t'=1}^{t} \left( \mathcal{Y}_{t'}(S_i) - \mathcal{Y}_{t'}(S_j) \right),$$

and the definition of regret is

$$\mathcal{R}(T, \pi) = \max_{S \in \mathcal{U}_\varepsilon} \sum_{t=1}^{T} \mathcal{Y}_t(S) - \mathbb{E}_\pi \left[ \sum_{t=1}^{T} R_t(S_t) \right]. \tag{4}$$

Here

$$\tilde{Y}_{i,t}(S) = \sum_{A \in \mathcal{K}^u} Y_{i,t}(A) \mathbb{P}(A_t = A \mid S), \quad \mathcal{Y}_t(S) = \frac{1}{N} \sum_{i \in \mathcal{U}} \tilde{Y}_{i,t}(S). \tag{5}$$

Given a fixed policy $\pi$, we provide the following hard instances. We define the first instance as $\nu_1 \in \mathcal{E}_0$, in which $Y_{i,t}(A) \sim \text{Bernoulli}(f_i(A))$. We denote the best arm as $S'$ and $S := \arg\min_{S \in \mathcal{U}_\varepsilon, S \neq S'} \tilde{\bar{\tau}}_T^1(S, S') \mathbb{E}_{\nu_1}[\mathcal{N}_S^T]$, where $\mathcal{N}_S^T = \sum_{t=1}^{T} \mathbf{1}\{S_t = S\}$ and $\tilde{\bar{\tau}}_T^1(S, S') := \frac{1}{N} \sum_{i \in \mathcal{U}} \sum_{A \in \mathcal{K}^u} f_i(A) \big( \mathbb{P}(A_t = A \mid S) - \mathbb{P}(A_t = A \mid S') \big)$. The difference in treatment effect between $S$ and $S'$,

$$\bar{\tau}_T^{\nu_1}(S, S') := \frac{1}{T} \frac{1}{N} \sum_{t=1}^{T} \sum_{i \in \mathcal{U}} \big( \tilde{Y}_{i,t}(S) - \tilde{Y}_{i,t}(S') \big),$$

can be equivalently expressed as (for brevity, we use $\bar{\tau}_1$ to denote $\bar{\tau}_T^{\nu_1}(S, S')$ in the subsequent discussion)

$$\bar{\tau}_1 = \frac{1}{T} \frac{1}{N} \sum_{t=1}^{T} \sum_{i \in \mathcal{U}} \sum_{A \in \mathcal{K}^u} Y_{i,t}(A) \Big( \mathbb{P}(A_t = A \mid S) - \mathbb{P}(A_t = A \mid S') \Big). \tag{6}$$

Based on the fact that $\frac{1}{N} \sum_{i \in \mathcal{U}} \sum_{A \in \mathcal{K}^u} Y_{i,t}(A) \big( \mathbb{P}(A_t = A \mid S) - \mathbb{P}(A_t = A \mid S') \big)$ is 1-sub-Gaussian, and for all $t \in [T]$

$$\mathbb{E}\left[ \frac{1}{N} \sum_{i \in \mathcal{U}} \sum_{A \in \mathcal{K}^u} Y_{i,t}(A) \Big( \mathbb{P}(A_t = A \mid S) - \mathbb{P}(A_t = A \mid S') \Big) \right] = \frac{1}{N} \sum_{i \in \mathcal{U}} \sum_{A \in \mathcal{K}^u} f_i(A) \Big( \mathbb{P}(A_t = A \mid S) - \mathbb{P}(A_t = A \mid S') \Big),$$

the Hoeffding inequality implies that, with probability at least $1 - \frac{1}{T}$,

$$\tilde{\bar{\tau}}_1 + \sqrt{\frac{2 \log(2T)}{T}} \geq \bar{\tau}_1. \tag{7}$$

On the other hand, we construct another instance as ($\beta \in (0, 1)$ is chosen as sufficiently small)

$$Y'_{i,t}(A) := \begin{cases} \text{Bernoulli}(f_i(A)) & \forall A \text{ satisfying } \mathbb{P}(A_t = A \mid S) = 0, \\ \text{Bernoulli}(f_i(A) - \beta) & \forall A \text{ satisfying } \mathbb{P}(A_t = A \mid S) > 0. \end{cases} \tag{8}$$

It leads to

$\bar{\tau}_2 = \bar{\tau}_{2,=0} + \bar{\tau}_{2,>0}$, where

$$\bar{\tau}_{2,=0} := \frac{1}{T} \frac{1}{N} \sum_{t=1}^{T} \sum_{i \in \mathcal{U}} \sum_{A \in \mathcal{K}^u} Y'_{i,t}(A) \Big( \mathbb{P}(A_t = A \mid S) - \mathbb{P}(A_t = A \mid S') \Big) \mathbf{1}\{\mathbb{P}(A_t = A \mid S) = 0\},$$

$$\bar{\tau}_{2,>0} := \frac{1}{T} \frac{1}{N} \sum_{t=1}^{T} \sum_{i \in \mathcal{U}} \sum_{A \in \mathcal{K}^u} Y'_{i,t}(A) \Big( \mathbb{P}(A_t = A \mid S) - \mathbb{P}(A_t = A \mid S') \Big) \mathbf{1}\{\mathbb{P}(A_t = A \mid S) > 0\}.$$

Follow the similar argument as Eq (7), we have with probability at least $1 - \frac{1}{T}$

$$\bar{\tau}_2 \geq \tilde{\bar{\tau}}_2 - \sqrt{\frac{2 \log(2T)}{T}}, \tag{9}$$

where

$\tilde{\bar{\tau}}_2 = \tilde{\bar{\tau}}_{2,=0} + \tilde{\bar{\tau}}_{2,>0}$, where

$$\tilde{\bar{\tau}}_{2,=0} := \frac{1}{N} \sum_{i \in \mathcal{U}} \sum_{A \in \mathcal{K}^u} f_i(A) \Big( \mathbb{P}(A_t = A \mid S) - \mathbb{P}(A_t = A \mid S') \Big) \mathbf{1}\{\mathbb{P}(A_t = A \mid S) = 0\},$$

$$\tilde{\bar{\tau}}_{2,>0} := \frac{1}{N} \sum_{i \in \mathcal{U}} \sum_{A \in \mathcal{K}^u} (f_i(A) - \beta) \Big( \mathbb{P}(A_t = A \mid S) - \mathbb{P}(A_t = A \mid S') \Big) \mathbf{1}\{\mathbb{P}(A_t = A \mid S) > 0\}.$$

Define event $\mathcal{E}_G := \{\tilde{\bar{\tau}}_1 + \sqrt{\frac{2\log(2T)}{T}} \geq \bar{\tau}_1, ; \bar{\tau}_2 \geq \tilde{\bar{\tau}}_2 - \sqrt{\frac{2\log(2T)}{T}}\}$. Under this event:

$$\bar{\tau}_2 - \bar{\tau}_1 \geq -2\sqrt{\frac{2\log(2T)}{T}} + \frac{1}{N}\sum_{i\in\mathcal{U}}\sum_{A\in\mathcal{K}^{\mathcal{U}}}(-\beta)\Big(\mathbb{P}(A_t = A \mid S) - \mathbb{P}(A_t = A \mid S')\Big)\mathbf{1}\{\mathbb{P}(A_t = A \mid S) > 0\}$$

$$= -2\sqrt{\frac{2\log(2T)}{T}} - \frac{1}{N}\sum_{i\in\mathcal{U}}\sum_{A\in\mathcal{K}^{\mathcal{U}}}\beta\Big(\mathbb{P}(A_t = A \mid S)\Big)\mathbf{1}\{\mathbb{P}(A_t = A \mid S) > 0\}$$

$$= -2\sqrt{\frac{2\log(2T)}{T}} - \beta.$$

On this basis, given any pre-specified estimator and strategy, which is recorded as $\{\hat{\Delta}_t\}_{t\in[T]}$, following Zhang and Wang [2024], Simchi-Levi and Wang [2024], we establish a hypothesis test as $\psi(\hat{\Delta}_T) = \arg\min_{i=1,2}|\hat{\Delta}_T - \bar{\tau}_i|$, implying that $\psi(\hat{\Delta}_T) \neq i, i \in \{1,2\}$ is a sufficient condition of $|\hat{\Delta}_T - \bar{\tau}_i| \geq \frac{1}{2}\beta + \sqrt{\frac{2\log(2T)}{T}}$. Therefore

$$\inf_{\hat{\Delta}_T}\max_{\nu\in\mathcal{E}_0}\mathbb{P}_\nu\left(|\hat{\Delta}_T - \bar{\tau}_\nu| \geq \frac{1}{2}\beta + \sqrt{\frac{2\log(2T)}{T}}\right) \geq \inf_{\hat{\Delta}_T}\max_{i\in\{1,2\}}\mathbb{P}_{\nu_i}\left(|\hat{\Delta}_T - \bar{\tau}_i| \geq \frac{1}{2}\beta + \sqrt{\frac{2\log(2T)}{T}}\right)$$

$$\geq \inf_{\hat{\Delta}_T}\max_{i\in\{1,2\}}\mathbb{P}_{\nu_i}\Big(\psi(\hat{\Delta}_T) \neq i\Big)$$

$$\geq \inf_{\psi}\max_{i\in\{1,2\}}\mathbb{P}_{\nu_i}(\psi \neq i). \tag{10}$$

The above equation can directly lead to

$$\text{RHS of (10)} \geq \frac{1}{2}(1 - \text{TV}(\mathbb{P}_{\nu_1}, \mathbb{P}_{\nu_2})) \geq \frac{1}{2}\left[1 - \sqrt{\frac{1}{2}\text{KL}(\mathbb{P}_{\nu_1}, \mathbb{P}_{\nu_2})}\right]. \tag{11}$$

Let $\mathbb{P}_{\nu,S}(\cdot)$ denotes the reward density distribution conditioning on arm $S$ in $\nu$. Due to the fact that $\text{KL}(\mathbb{P}_{\nu_1}, \mathbb{P}_{\nu_2}) = \mathbb{E}_{\nu_1}[\mathcal{N}_S^T]\text{KL}(\mathbb{P}_{\nu_1,S}(\cdot), \mathbb{P}_{\nu_2,S}(\cdot))$, and

$$\text{KL}(\mathbb{P}_{\nu_1,S}(\cdot), \mathbb{P}_{\nu_2,S}(\cdot)) = \int_X p_{\nu_1,S}(X)log\left(\frac{p_{\nu_1,S}(X)}{p_{\nu_2,S}(X)}\right)dX \leq q\beta^2 N, \tag{12}$$

where $q > 0$ is a constant. It achieves that

$$\text{KL}(\mathbb{P}_{\nu_1}, \mathbb{P}_{\nu_2}) = \mathbb{E}_{\nu_1}[\mathcal{N}_S^T]\text{KL}(\mathbb{P}_{\nu_1,S}(\cdot), \mathbb{P}_{\nu_2,S}(\cdot))$$

$$\leq q\beta^2 N\mathbb{E}_{\nu_1}[\mathcal{N}_S^T]$$

$$\leq q\beta^2 N\frac{\mathcal{R}_{\nu_1}^{stoc}(T,\pi)}{|\mathcal{U}_\mathcal{E}||\tilde{\bar{\tau}}_1|} \tag{13}$$

$$\leq q\left(\beta + 2\sqrt{\frac{2\log(2T)}{T}}\right)^2 N\frac{\mathcal{R}_{\nu_1}^{stoc}(T,\pi)}{|\mathcal{U}_\mathcal{E}||\tilde{\bar{\tau}}_1|},$$

where $\mathcal{R}_{\nu_1}^{stoc}(\cdot)$ denotes the regret defined in the stochastic bandit setting under instance $\nu_1$. Here the last inequality is due to the definition of $S$. Combining (11)-(13), it implies

$$\inf_{\hat{\Delta}_T}\max_{\nu\in\mathcal{E}_0}\mathbb{P}_\nu\left(\max_{S_i,S_j\in\mathcal{U}_\mathcal{E}}|\hat{\Delta}_T(S_i,S_j) - \bar{\tau}_T^\nu(S_i,S_j)| \geq \frac{\beta}{2} + \sqrt{\frac{2\log(2T)}{T}}\right)$$

$$\geq \frac{1}{2}\left[1 - \sqrt{\frac{1}{2}q\left(\beta + 2\sqrt{\frac{2\log(2T)}{T}}\right)^2 N\frac{\mathcal{R}_{\nu_1}^{stoc}(T,\pi)}{|\mathcal{U}_\mathcal{E}||\tilde{\bar{\tau}}_1|}}\right]. \tag{14}$$

Moreover, we aim to relate the regret in adversarial and stochastic settings. For any feasible stochastic instance $\nu$, obtained for example by Bernoulli sampling of $Y_{i,t}(A)$, we have

$$\mathcal{R}_\nu(T,\pi) \geq \mathcal{R}_\nu^{stoc}(T,\pi), \tag{15}$$

where the inequality follows from Jensen's inequality. Combining (14)-(15), we get under event $\mathcal{E}_G$

$$
\inf_{\hat{\Delta}_T} \max_{\nu \in \mathcal{E}_0} \mathbb{P}_\nu \left( \max_{S_i, S_j \in \mathcal{U}_\mathcal{E}} |\hat{\Delta}_T(S_i, S_j) - \bar{\tau}_T^\nu(S_i, S_j)| \geq \frac{\beta}{2} + \sqrt{\frac{2\log(2T)}{T}} \right)
$$
$$
\geq \frac{1}{2} \left[ 1 - \sqrt{\frac{1}{2} q \left( \beta + 2\sqrt{\frac{2\log(2T)}{T}} \right)^2 N \frac{\mathcal{R}_{\nu_1}(T, \pi)}{|\mathcal{U}_\mathcal{E}||\tilde{\bar{\tau}}_1|}} \right]. \tag{16}
$$

As a consequence,

$$
\inf_{\hat{\Delta}_T} \max_{\nu \in \mathcal{E}_0} \mathbb{E}_\nu \left( \max_{S_i, S_j \in \mathcal{U}_\mathcal{E}} |\hat{\Delta}_T(S_i, S_j) - \bar{\tau}_T^\nu(S_i, S_j)| \right) \sqrt{\mathcal{R}_{\nu_1}(T, \pi)}
$$
$$
\geq \frac{1}{2} \left( 1 - \frac{2}{T} \right) \left( \frac{\beta}{2} + \sqrt{\frac{2\log(2T)}{T}} \right) \left[ 1 - \sqrt{\frac{1}{2} q \left( \beta + 2\sqrt{\frac{2\log(2T)}{T}} \right)^2 N \frac{\mathcal{R}_{\nu_1}(T, \pi)}{|\mathcal{U}_\mathcal{E}||\tilde{\bar{\tau}}_1|}} \right] \sqrt{\mathcal{R}_{\nu_1}(T, \pi)}. \tag{17}
$$

When we choose $\beta$ such that $q \left( \beta + 2\sqrt{\frac{\log(T/2)}{2T}} \right)^2 N \frac{\mathcal{R}_{\nu_1}(T, \pi)}{|\mathcal{U}_\mathcal{E}||\tilde{\bar{\tau}}_1|} = \frac{1}{2}$, it follows

$$
(17) = \frac{1}{8} \left( 1 - \frac{2}{T} \right) \sqrt{\frac{|\mathcal{U}_\mathcal{E}||\tilde{\bar{\tau}}_1|}{2qN}} = \Omega_{K,T}(\sqrt{|\mathcal{U}_\mathcal{E}|}). \tag{18}
$$

$\square$

## D  Proof of Theorem 5.5

The following lemma is important in the proof of Theorem 5.5:

**Lemma D.1.** *Following the setting in Theorem 5.5, for all $S_i, S_j \in \mathcal{U}_\mathcal{E}$, the sequence $\{\hat{\tau}_t(S_i, S_j)\}_{t=1}^\infty$ satisfies the Lindeberg-type uniform integrability condition (Condition L2 of Proposition 2.5) outlined by Waudby-Smith et al. [2021], i.e., there exists $\beta \in (0, 1)$ such that*

$$
\sum_{t=1}^\infty \frac{\mathbb{E}\left[ \left( \hat{\tau}_t(S_i, S_j) - \tau_t(S_i, S_j) \right)^2 \mathbf{1}\left\{ \left( \hat{\tau}_t(S_i, S_j) - \tau_t(S_i, S_j) \right)^2 > \left( \mathcal{V}_t(S_i, S_j) \right)^\beta \right\} \right]}{\left( \mathcal{V}_t(S_i, S_j) \right)^\beta} < \infty \quad a.s.,
$$

*where $\mathcal{V}_t(S_i, S_j) = \sum_{t'=1}^t \mathbf{V}\left( \hat{\tau}_{t'}(S_i, S_j) \mid \mathcal{F}_{t'} \right)$ is the cumulative conditional variance.*

*Proof of Lemma D.1.* We first upper bound $\left( \hat{\tau}_t(S_i, S_j) - \tau_t(S_i, S_j) \right)^2$. Based on the definition of our IPW estimator, we have

$$
\left( \hat{\tau}_t(S_i, S_j) - \tau_t(S_i, S_j) \right)^2
$$
$$
= \left( \frac{\mathbf{1}\{S_t = S_i\} R_t(S_i)}{\pi_t^{\text{MAD}}(S_i)} - \frac{\mathbf{1}\{S_t = S_j\} R_t(S_j)}{\pi_t^{\text{MAD}}(S_j)} - \tau_t(S_i, S_j) \right)^2
$$
$$
\leq \frac{4}{\left( \pi_t^{\text{MAD}}(S_i) \wedge \pi_t^{\text{MAD}}(S_j) \right)^2} + \frac{8}{\left( \pi_t^{\text{MAD}}(S_i) \wedge \pi_t^{\text{MAD}}(S_j) \right)} + 4
$$
$$
\leq \frac{16}{\left( \pi_t^{\text{MAD}}(S_i) \wedge \pi_t^{\text{MAD}}(S_j) \right)^2},
$$

where the first inequality is due to $R_t(S) \in [0, 1]$. Note that, based on the setup of Theorem 5.5, we have $\frac{1}{(\pi_t^{\text{MAD}}(S))^2} = O(t^{2\alpha})$ for all $S \in \mathcal{U}_\mathcal{E}$. This implies that $\left( \hat{\tau}_t(S_i, S_j) - \tau_t(S_i, S_j) \right)^2 = O(t^{2\alpha})$. Furthermore, based on Assumption 5.3, we have $\mathcal{V}_t(S_i, S_j) = \Omega(t)$. Therefore, by setting $\beta \in \left( 2\alpha, 1 \right)$, there always exists a finite time $t'$ such that for all $t \geq t'$, $\left( \hat{\tau}_t(S_i, S_j) - \tau_t(S_i, S_j) \right)^2 \leq$

$\big(\mathcal{V}_t(S_i, S_j)\big)^\beta$, and

$$\sum_{t=1}^{\infty} \frac{\mathbb{E}\big[\big(\hat{\tau}_t(S_i, S_j) - \tau_t(S_i, S_j)\big)^2 \mathbf{1}\big\{\big(\hat{\tau}_t(S_i, S_j) - \tau_t(S_i, S_j)\big)^2 > \big(\mathcal{V}_t(S_i, S_j)\big)^\beta\big\}\big]}{\big(\mathcal{V}_t(S_i, S_j)\big)^\beta}$$
$$= \sum_{t=1}^{t'} \frac{\mathbb{E}\big[\big(\hat{\tau}_t(S_i, S_j) - \tau_t(S_i, S_j)\big)^2 \mathbf{1}\big\{\big(\hat{\tau}_t(S_i, S_j) - \tau_t(S_i, S_j)\big)^2 > \big(\mathcal{V}_t(S_i, S_j)\big)^\beta\big\}\big]}{\big(\mathcal{V}_t(S_i, S_j)\big)^\beta}$$
$$< \infty \quad \text{a.s.}$$

Here we finish the proof of Lemma D.1. $\qquad\square$

Based on Lemma D.1, we can prove Theorem 5.5.

*Proof of Theorem 5.5.* Based on Assumption 5.3, Lemma D.1, and Proposition 2.5 in Waudby-Smith et al. [2021], $\{\hat{\hat{\tau}}_t(S_i, S_j) \pm C_t(S_i, S_j)\}_{t=1}^{\infty}$ constitutes an asymptotic $(1 - \tilde{\delta})$ CS, where

$$C_t(S_i, S_j) = \sqrt{\frac{2\big(\mathcal{V}_t(S_i, S_j)\eta^2 + 1\big)}{t^2\eta^2} \log\bigg(\frac{\sqrt{\mathcal{V}_t(S_i, S_j)\eta^2 + 1}}{\tilde{\delta}}\bigg)}.$$

Besides, based on the definition of the variance, we know that $\mathcal{V}_t \leq \tilde{\mathcal{V}}_t$, where $\tilde{\mathcal{V}}_t = \sum_{t'=1}^{t} \sigma_{t'}^2(S_i, S_j) = \sum_{t'=1}^{t} \Big(\frac{(\mathcal{Y}_t(S_i))^2}{\pi_{t'}^{\text{MAD}}(S_i)} + \frac{(\mathcal{Y}_t(S_j))^2}{\pi_{t'}^{\text{MAD}}(S_j)}\Big)$. Therefore, $\{\hat{\hat{\tau}}_t(S_i, S_j) \pm \tilde{C}_t(S_i, S_j)\}_{t=1}^{\infty}$ is also an asymptotic $(1 - \tilde{\delta})$ CS, where

$$C_t(S_i, S_j) \leq \tilde{C}_t(S_i, S_j) = \sqrt{\frac{2\big(\tilde{\mathcal{V}}_t(S_i, S_j)\eta^2 + 1\big)}{t^2\eta^2} \log\bigg(\frac{\sqrt{\tilde{\mathcal{V}}_t(S_i, S_j)\eta^2 + 1}}{\tilde{\delta}}\bigg)}.$$

Define $\hat{\sigma}_t^2(S_i, S_j) = \Big(\frac{1}{\pi_t^{\text{MAD}}(S_i)} + \frac{1}{\pi_t^{\text{MAD}}(S_j)}\Big)$ as the estimator of $\sigma_t^2(S_i, S_j)$, and let $\hat{\mathcal{V}}_t = \sum_{t'=1}^{t} \hat{\sigma}_{t'}^2(S_i, S_j)$. Since $\hat{\mathcal{V}}_t \geq \tilde{\mathcal{V}}_t$, the sequence $\{\hat{\hat{\tau}}_t(S_i, S_j) \pm \hat{C}_t(S_i, S_j)\}_{t=1}^{\infty}$ forms an asymptotic $(1 - \tilde{\delta})$ confidence sequence, where

$$\hat{C}_t(S_i, S_j) = \sqrt{\frac{2\big(\hat{\mathcal{V}}_t(S_i, S_j)\eta^2 + 1\big)}{t^2\eta^2} \log\bigg(\frac{\sqrt{\hat{\mathcal{V}}_t(S_i, S_j)\eta^2 + 1}}{\tilde{\delta}}\bigg)}.$$

We finally show that $\hat{C}_t(S_i, S_j) = \tilde{O}\big(|\mathcal{U}_\mathcal{E}|^{\frac{1}{2}} t^{\frac{\alpha-1}{2}}\big)$ for all $S_i, S_j \in \mathcal{U}_\mathcal{E}$. We first upper bound $\hat{\mathcal{V}}_t(S_i, S_j)$, i.e.,

$$\hat{\mathcal{V}}_t(S_i, S_j) = \sum_{t'=1}^{t} \bigg(\frac{1}{\pi_t^{\text{MAD}}(S_i)} + \frac{1}{\pi_t^{\text{MAD}}(S_j)}\bigg)$$
$$\leq \sum_{t'=1}^{t} \big(2|\mathcal{U}_\mathcal{E}|t'^\alpha\big)$$
$$= O\big(|\mathcal{U}_\mathcal{E}|t^{1+\alpha}\big).$$

Then

$$\hat{C}_t(S_i, S_j) = O\Bigg(\sqrt{\frac{\big(|\mathcal{U}_\mathcal{E}|t^{1+\alpha}\eta^2 + 1\big)}{t^2\eta^2} \log\bigg(\frac{\sqrt{|\mathcal{U}_\mathcal{E}|t^{1+\alpha}\eta^2 + 1}}{\tilde{\delta}}\bigg)}\Bigg) = \tilde{O}\big(|\mathcal{U}_\mathcal{E}|^{\frac{1}{2}} t^{\frac{\alpha-1}{2}}\big),$$

and it will converge to 0 when $t \to \infty$. This concludes the proof of Theorem 5.5. $\qquad\square$

# E Proof of Theorem 5.6

*Proof of Theorem 5.6, Claim (i).* Based on the result in Theorem 5.5, for all $S_i \neq S_j$, with probability at least $1 - \tilde{\delta}$, we have

$$
\begin{aligned}
|\hat{\Delta}^{(i,j)} - \Delta^{(i,j)}| &\leq 2\hat{C}_T(S_i, S_j) \\
&= 2\sqrt{\frac{2(\hat{\mathcal{V}}_T(S_i, S_j)\eta^2 + 1)}{T^2\eta^2} \log\left(\frac{\sqrt{\hat{\mathcal{V}}_T(S_i, S_j)\eta^2 + 1}}{\tilde{\delta}}\right)} \\
&= \tilde{O}\left(|\mathcal{U}_{\mathcal{E}}|T^{\alpha - \frac{1}{2}}\right),
\end{aligned}
$$

where the first inequality is owing to Theorem 5.5, and the last inequality is owing to the definition of $\hat{\mathcal{V}}_T(S_i, S_j)$. Finally, set $\tilde{\delta} = 1/T$, we have

$$
\mathbb{E}[|\hat{\Delta}^{(i,j)} - \Delta^{(i,j)}|] \leq 2(1 - \tilde{\delta})\hat{C}_t(S_i, S_j) + \tilde{\delta}T = \tilde{O}\left(|\mathcal{U}_{\mathcal{E}}|T^{\alpha - \frac{1}{2}}\right).
$$

$\square$

*Proof of Theorem 5.6 Claim (ii).* Let $\mathbb{E}_{\text{MAD}}[\cdot]$ and $\mathbb{E}_{\text{ALG}}[\cdot]$ denote the expectations taken with respect to the MAD and ALG (EXP3), respectively. Recall the definition of regret:

$$
\mathcal{R}(T, \pi^{\text{MAD}}) = \max_{S \in \mathcal{U}_{\mathcal{E}}} \sum_{t=1}^{T} \mathcal{Y}_t(S) - \mathbb{E}_{\text{MAD}}\left[\sum_{t=1}^{T} R_t(S_t)\right], \tag{19}
$$

we also define

$$
\mathcal{R}(T, \pi^{\text{MAD}}, i) = \sum_{t=1}^{T} \mathcal{Y}_t(S_i) - \mathbb{E}_{\text{MAD}}\left[\sum_{t=1}^{T} R_t(S_t)\right]. \tag{20}
$$

As the "regret" assuming a fixed super arm $S_i$ is optimal for all $T$ rounds, while $\mathcal{R}(T, \pi)$ measures the actual regret relative to the best super arm at each round. If we can establish that $\mathcal{R}(T, \pi^{\text{MAD}}, i) = \tilde{O}(\sqrt{|\mathcal{U}_{\mathcal{E}}|T} + T^{1-\alpha})$ for all $S_i \in \mathcal{U}_{\mathcal{E}}$, it follows directly that $\mathcal{R}(T, \pi^{\text{MAD}}) = \tilde{O}(\sqrt{|\mathcal{U}_{\mathcal{E}}|T} + T^{1-\alpha})$.

Based on the definition of the MAD, we can decompose Eq (20) as

$$
\begin{aligned}
\mathcal{R}(T, \pi^{\text{MAD}}, i) &= \sum_{t=1}^{T} \mathcal{Y}_t(S_i) - \mathbb{E}_{\text{MAD}}\left[\sum_{t=1}^{T} R_t(S_t)\right] \\
&= \sum_{t=1}^{T} \mathcal{Y}_t(S_i) - \sum_{t=1}^{T}\left(\delta_t\left(\frac{\sum_{S \in \mathcal{U}_{\mathcal{E}}} \mathcal{Y}_t(S)}{|\mathcal{U}_{\mathcal{E}}|}\right) + (1 - \delta_t)\mathbb{E}_{\text{ALG}}[R_t(S_t)]\right) \\
&= \mathcal{R}(T, \pi^{\text{ALG}}, i) + \sum_{t=1}^{T} \delta_t\left(\mathbb{E}_{\text{ALG}}[R_t(S_t)] - \frac{\sum_{S \in \mathcal{U}_{\mathcal{E}}} \mathcal{Y}_t(S)}{|\mathcal{U}_{\mathcal{E}}|}\right) \\
&\leq \mathcal{R}(T, \pi^{\text{ALG}}, i) + 2T^{1-\alpha},
\end{aligned} \tag{21}
$$

where the third inequality is owing to the definition that $\mathcal{R}(T, \pi^{\text{ALG}}, i) = \sum_{t=1}^{T} \mathcal{Y}_t(S_i) - \sum_{t=1}^{T} \mathbb{E}_{\text{ALG}}[R_t(S_t)]$. We further decompose $\mathcal{R}(T, \pi^{\text{ALG}}, i)$. Let $M$ be such that $T \in \mathcal{L}_M$, and define $\mathcal{R}(\mathcal{L}_m, \pi^{\text{ALG}}, i)$ as $\mathcal{R}(\mathcal{L}_m, \pi^{\text{ALG}}, i) = \sum_{t \in \mathcal{L}_m} \mathcal{Y}_t(S_i) - \mathbb{E}_{\text{MAD}}\left[\sum_{t \in \mathcal{L}_m} R_t(S_t)\right]$. It follows directly that $\mathcal{R}(T, \pi^{\text{ALG}}, i) \leq \sum_{m=1}^{M} \mathcal{R}(\mathcal{L}_m, \pi^{\text{ALG}}, i)$.

We now focusing on upper bound $\mathcal{R}(\mathcal{L}_m, \pi^{\text{ALG}}, i)$. Set $\hat{R}_{\mathcal{L}_m, t_m-1}(S) = 0$ for all $S \in \mathcal{U}_\mathcal{E}$. Based on the unbiasedness of the IPW estimator, we have:

$$\mathbb{E}_{\text{ALG}}[\hat{R}_{\mathcal{L}_m, t+2^{m-1}-1}(S)] = \sum_{t \in \mathcal{L}_m} \mathcal{Y}_t(S), \, \forall S \in \mathcal{U}_\mathcal{E}, \text{and}$$

$$\mathbb{E}_{\text{ALG}}\Big[R_t(S_t)\Big|\mathcal{H}_{t-1}\Big] = \sum_{S \in \mathcal{U}_\mathcal{E}} \pi_t^{\text{ALG}}(S)\mathcal{Y}_t(S)$$

$$= \sum_{S \in \mathcal{U}_\mathcal{E}} \pi_t^{\text{ALG}}(S)\mathbb{E}_{\text{ALG}}\Big[\hat{R}_{\mathcal{L}_m, t}(S) - \hat{R}_{\mathcal{L}_m, t-1}(S)\Big|\mathcal{H}_{t-1}\Big], \, \forall t \in \mathcal{L}_m.$$

(22)

According to Eq (22), Eq (20) can be rewritten as

$$\mathcal{R}(\mathcal{L}_m, \pi^{\text{ALG}}, i) = \mathbb{E}_{\text{ALG}}[\hat{R}_{\mathcal{L}_m, t_m+2^{m-1}-1}(S_i)] - \mathbb{E}_{\text{ALG}}\Big[\sum_{t \in \mathcal{L}_m} R_t(S_t)\Big]$$

$$= \mathbb{E}_{\text{ALG}}[\hat{R}_{\mathcal{L}_m, t_m+2^{m-1}-1}(S_i)] - \mathbb{E}_{\text{ALG}}\Big[\mathbb{E}_{\text{ALG}}\Big[\sum_{t \in \mathcal{L}_m} R_t(S_t)\Big|\mathcal{H}_{t-1}\Big]\Big]$$

$$= \mathbb{E}_{\text{ALG}}[\hat{R}_{\mathcal{L}_m, t_m+2^{m-1}-1}(S_i)] - \mathbb{E}_{\text{ALG}}\Big[\sum_{t \in \mathcal{L}_m}\sum_{S \in \mathcal{U}_\mathcal{E}} \pi_t^{\text{ALG}}(S)\mathbb{E}_{\text{ALG}}\Big[\Big(\hat{R}_{\mathcal{L}_m, t}(S) - \hat{R}_{\mathcal{L}_m, t-1}(S)\Big)\Big|\mathcal{H}_{t-1}\Big]\Big]$$

$$= \mathbb{E}_{\text{ALG}}\Big[\hat{R}_{\mathcal{L}_m, t_m+2^{m-1}-1}(S_i) - \sum_{t=1}^{T}\sum_{S \in \mathcal{U}_\mathcal{E}} \pi_t^{\text{ALG}}(S)\Big(\hat{R}_{\mathcal{L}_m, t}(S) - \hat{R}_{\mathcal{L}_m, t-1}(S)\Big)\Big]$$

$$= \mathbb{E}_{\text{ALG}}\Big[\hat{R}_{\mathcal{L}_m, t_m+2^{m-1}-1}(S_i) - \hat{R}_{\mathcal{L}_m}\Big],$$

where the first and third equalities follow from the tower rule, while the last equality holds due to our definition: $\hat{R}_{\mathcal{L}_m} = \sum_{t \in \mathcal{L}_m}\sum_{S \in \mathcal{U}_\mathcal{E}} \pi_t^{\text{ALG}}(S)\big(\hat{R}_{\mathcal{L}_m, t}(S) - \hat{R}_{\mathcal{L}_m, t-1}(S)\big)$.

For $t \in \mathcal{L}_m$, we define $W_t = \sum_{S \in \mathcal{U}_\mathcal{E}} \exp\big(\epsilon_m \hat{R}_{\mathcal{L}_m, t}(S)\big)$. Consider the ratio between successive $W_t$ and $W_{t-1}$: $\frac{W_t}{W_{t-1}}$. Using the definition of:

$$\pi_t^{\text{ALG}}(S) = \frac{\exp\big(\epsilon_m \hat{R}_{\mathcal{L}_m, t-1}(S)\big)}{W_{t-1}}.$$

(23)

We rewrite the ratio as:

$$\frac{W_t}{W_{t-1}} = \sum_{S \in \mathcal{U}_\mathcal{E}} \pi_t^{\text{ALG}}(S) \exp\Big(\epsilon_m\big(\hat{R}_{\mathcal{L}_m, t}(S) - \hat{R}_{\mathcal{L}_m, t-1}(S)\big)\Big).$$

(24)

We now introduce two inequalities: 1) $\exp(x) \leq 1 + x + x^2, \forall x \leq 1$ ; 2) $1 + x \leq \exp(x), \forall x > 0$. Based on these two inequalities, we can rewrite Eq (24) as:

$$\sum_{S \in \mathcal{U}_\mathcal{E}} \pi_t^{\text{ALG}}(S) \exp\Big(\epsilon_m\big(\hat{R}_{\mathcal{L}_m, t}(S) - \hat{R}_{\mathcal{L}_m, t-1}(S)\big)\Big)$$

$$\leq \Big(1 + \epsilon_m \sum_{S \in \mathcal{U}_\mathcal{E}} \pi_t^{\text{ALG}}(S)\Big(\hat{R}_{\mathcal{L}_m, t}(S) - \hat{R}_{\mathcal{L}_m, t-1}(S)\Big) + \epsilon_m^2 \sum_{S \in \mathcal{U}_\mathcal{E}} \pi_t^{\text{ALG}}(S)\Big(\hat{R}_{\mathcal{L}_m, t}(S) - \hat{R}_{\mathcal{L}_m, t-1}(S)\Big)^2\Big)$$

$$\leq \exp\Big(\epsilon_m \sum_{S \in \mathcal{U}_\mathcal{E}} \pi_t^{\text{ALG}}(S)\Big(\hat{R}_{\mathcal{L}_m, t}(S) - \hat{R}_{\mathcal{L}_m, t-1}(S)\Big) + \epsilon_m^2 \sum_{S \in \mathcal{U}_\mathcal{E}} \pi_t^{\text{ALG}}(S)\Big(\hat{R}_{\mathcal{L}_m, t}(S) - \hat{R}_{\mathcal{L}_m, t-1}(S)\Big)^2\Big).$$

Multiplying these ratios from $t_m$ to $t_m + 2^{m-1} - 1$, we obtain:

$$W_{t_m+2^{m-1}-1} = |\mathcal{U}_\mathcal{E}| \prod_{t \in \mathcal{L}_m} \frac{W_t}{W_{t-1}} \leq |\mathcal{U}_\mathcal{E}| \exp\Big(\epsilon_m \hat{R}_{\mathcal{L}_m} + \epsilon_m^2 \sum_{t \in \mathcal{L}_m}\sum_{S \in \mathcal{U}_\mathcal{E}} \pi_t^{\text{ALG}}(S)\Big(\hat{R}_{\mathcal{L}_m, t}(S) - \hat{R}_{\mathcal{L}_m, t-1}(S)\Big)^2\Big).$$

Taking logarithms and rearranging the above equation, it yields:

$$\hat{R}_{\mathcal{L}_m, t_m + 2^{m-1} - 1}(S_i) - \hat{R}_{\mathcal{L}_m} \leq \frac{\log\left(|\mathcal{U}_{\mathcal{E}}|\right)}{\epsilon_m} + \epsilon_m \sum_{t \in \mathcal{L}_m} \sum_{S \in \mathcal{U}_{\mathcal{E}}} \pi_t^{\text{ALG}}(S)\left(\hat{R}_{\mathcal{L}_m, t}(S) - \hat{R}_{\mathcal{L}_m, t-1}(S)\right)^2.$$

Recalling the definition $\mathcal{R}(\mathcal{L}_m, \pi^{\text{ALG}}, i) = \mathbb{E}_{\text{ALG}}\left[\hat{R}_{\mathcal{L}_m, t_m + 2^{m-1} - 1}(S_i) - \hat{R}_{\mathcal{L}_m}\right]$, we obtain:

$$\mathcal{R}(\mathcal{L}_m, \pi^{\text{ALG}}, i) \leq \frac{\log\left(|\mathcal{U}_{\mathcal{E}}|\right)}{\epsilon_m} + \mathbb{E}_{\text{ALG}}\left[\epsilon_m \sum_{t \in \mathcal{L}_m} \sum_{S \in \mathcal{U}_{\mathcal{E}}} \pi_t^{\text{ALG}}(S)\left(\hat{R}_{\mathcal{L}_m, t}(S) - \hat{R}_{\mathcal{L}_m, t-1}(S)\right)^2\right].$$

We then try to bound $\mathbb{E}_{\text{ALG}}\left[\epsilon_m \sum_{t \in \mathcal{L}_m} \sum_{S \in \mathcal{U}_{\mathcal{E}}} \pi_t^{\text{ALG}}(S)\left(\hat{R}_{\mathcal{L}_m, t}(S) - \hat{R}_{\mathcal{L}_m, t-1}(S)\right)^2\right]$, there is

$$\mathbb{E}_{\text{ALG}}\left[\epsilon_m \sum_{t \in \mathcal{L}_m} \sum_{S \in \mathcal{U}_{\mathcal{E}}} \pi_t^{\text{ALG}}(S)\left(\hat{R}_{\mathcal{L}_m, t}(S) - \hat{R}_{\mathcal{L}_m, t-1}(S)\right)^2\right]$$

$$=\mathbb{E}_{\text{ALG}}\left[\epsilon_m \sum_{t \in \mathcal{L}_m} \sum_{S \in \mathcal{U}_{\mathcal{E}}} \pi_t^{\text{ALG}}(S)\left(1 - \frac{\mathbf{1}\{S_t = S\}\left(1 - R_t(S)\right)}{\pi_t^{\text{ALG}}(S)}\right)^2\right]$$

$$=\mathbb{E}_{\text{ALG}}\left[\epsilon_m \sum_{t \in \mathcal{L}_m} \sum_{S \in \mathcal{U}_{\mathcal{E}}} \pi_t^{\text{ALG}}(S)\left(1 - \frac{2 \times \mathbf{1}\{S_t = S\}\left(1 - R_t(S)\right)}{\pi_t^{\text{ALG}}(S)} + \frac{\mathbf{1}\{S_t = S\}\left(1 - R_t(S)\right)^2}{\pi_t^{\text{ALG}}(S)^2}\right)\right]$$

$$=\mathbb{E}_{\text{ALG}}\left[\epsilon_m \sum_{t \in \mathcal{L}_m} \left(2R_t(S_t)\right)\right] + \mathbb{E}_{\text{ALG}}\left[\epsilon_m \sum_{t \in \mathcal{L}_m} \sum_{S \in \mathcal{U}_{\mathcal{E}}} \pi_t^{\text{ALG}}(S)\left(\frac{\mathbf{1}\{S_t = S\}\left(1 - R_t(S)\right)^2}{\pi_t^{\text{ALG}}(S)^2}\right)\Big|\mathcal{H}_{t-1}\right]$$

$$=\mathbb{E}_{\text{ALG}}\left[\epsilon_m \sum_{t \in \mathcal{L}_m} \left(2R_t(S_t) - 1\right) + \epsilon_m \sum_{t \in \mathcal{L}_m} \sum_{S \in \mathcal{U}_{\mathcal{E}}} \left(1 - R_t(S)\right)^2\right]$$

$$\leq|\mathcal{U}_{\mathcal{E}}|2^{m-1}\epsilon_m.$$

Based on the definition of $\epsilon_m$, we conclude that $\mathcal{R}(\mathcal{L}_m, \pi^{\text{ALG}}, i) = \tilde{O}(\sqrt{|\mathcal{U}_{\mathcal{E}}|2^{m-1}})$. We can upper bound $\mathcal{R}(T, \pi^{\text{ALG}}, i)$ by $\sum_{m=1}^{M} \mathcal{R}(\mathcal{L}_m, \pi^{\text{ALG}}, i) = \tilde{O}\left(\sum_{m=1}^{M} \sqrt{|\mathcal{U}_{\mathcal{E}}|2^{m-1}}\right) = \tilde{O}(\sqrt{|\mathcal{U}_{\mathcal{E}}|}2^{M/2})$. Owing to $M \leq \log_2(T) + 1$, we have $\mathcal{R}(T, \pi^{\text{ALG}}, i) = \tilde{O}(\sqrt{|\mathcal{U}_{\mathcal{E}}|T})$. We can finally bound $\mathcal{R}(T, \pi^{\text{MAD}}, i)$ and $\mathcal{R}(T, \pi^{\text{MAD}})$ by $\tilde{O}(\sqrt{|\mathcal{U}_{\mathcal{E}}|T} + T^{1-\alpha})$. $\qquad\square$

## F  Additional Experimental Results

In this section, we present four additional experiment instances along with the corresponding results.

**Instance 1: single unit.**  In this setup, the network consists of a single unit (as illustrated in Fig. 3(a)), making it identical to the case considered in Liang and Bojinov [2023]. Additionally, the action set is defined to include five actions, i.e., $\mathcal{A} = \{0, \ldots, 4\}$. The reward structure and baseline algorithms are configured in the same manner as described in Section 6.

**Instance 2: 6 units.**  The network in this setup consists of 6 units, organized in a loop topology, as illustrated in Fig. 4(a). Furthermore, the network is divided into three clusters, with the cluster structure also depicted in Fig. 4(a). The configuration of the action set, reward structure, and baseline algorithms follows the same setup as described in Section 6.

**Instance 3: 10 units case 1**  In this setup, the network consists of 10 units, with the topology structure depicted in Fig. 5(a). Additionally, we divide the network into three clusters, and the cluster structure is also illustrated in Fig. 5(a). The configuration of the action set, reward structure, and baseline algorithms remains the same as described in Section 6.

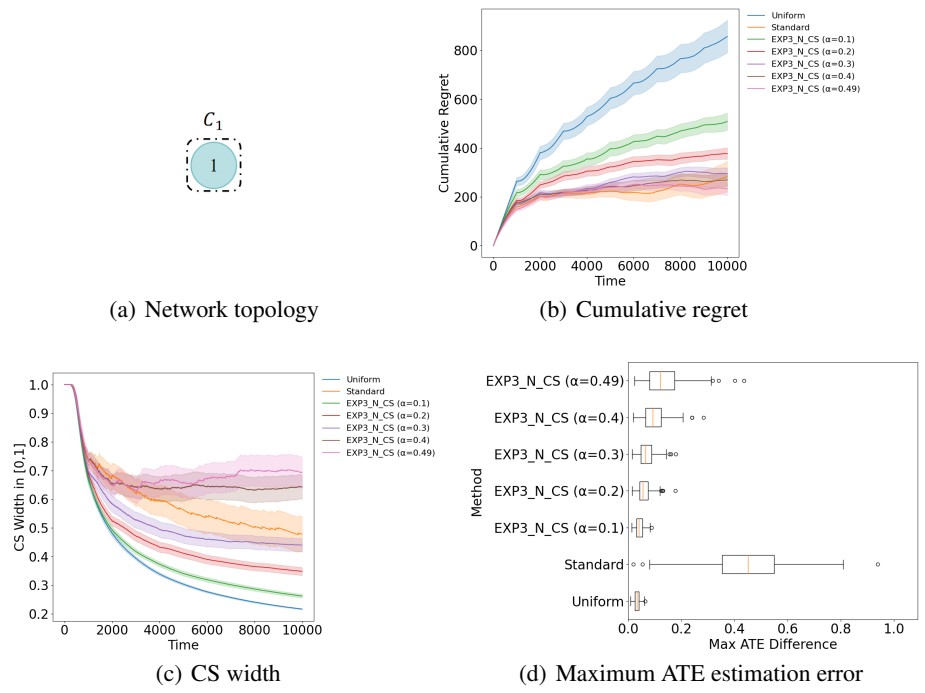

(a) Network topology

(b) Cumulative regret

(c) CS width

(d) Maximum ATE estimation error

Figure 3: Experimental results of instance 1.

**Instance 4: 10 units case 2** The network consists of 10 units arranged in a star-like topology, as shown in Figure 6(a). At the center of the network lies a single unit forming the central cluster, which is directly connected to every unit in three outer clusters. Each of these outer clusters comprises 3 units, resulting in a total of 9 peripheral units connected to the central cluster. The configuration of the action set, reward structure, and baseline algorithms remains the same as described in Section 6.

We ran the algorithms 100 times and reported the average results.

The experimental results are presented in Fig. 3, 4, 5 and 6. For cumulative regret, both the Standard approach and `EXP3-N-CS` with larger $\alpha$ values achieve the lowest regret, while the Uniform baseline incurs the highest regret. For continual inference, although Standard exhibits narrower CS widths than some `EXP3-N-CS` variants, its intervals are invalid due to the lack of theoretical guarantees. This issue is reflected in the maximum ATE estimation error, where Standard exhibits the largest errors with many outliers. In contrast, `EXP3-N-CS` with moderate or large $\alpha$ and the *Uniform* baseline achieve lower estimation errors.

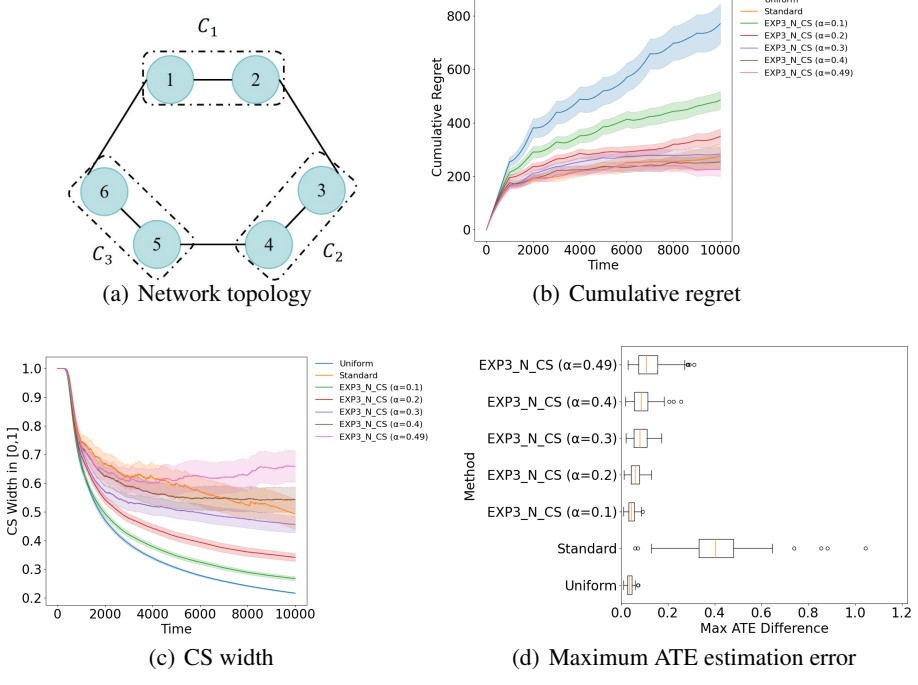

(a) Network topology          (b) Cumulative regret

(c) CS width          (d) Maximum ATE estimation error

Figure 4: Experimental results of instance 2.


Figure 5: Experimental results of instance 3.

- The paper should point out any strong assumptions and how robust the results are to violations of these assumptions (e.g., independence assumptions, noiseless settings, model well-specification, asymptotic approximations only holding locally). The authors should reflect on how these assumptions might be violated in practice and what the implications would be.
- The authors should reflect on the scope of the claims made, e.g., if the approach was only tested on a few datasets or with a few runs. In general, empirical results often depend on implicit assumptions, which should be articulated.
- The authors should reflect on the factors that influence the performance of the approach. For example, a facial recognition algorithm may perform poorly when image resolution is low or images are taken in low lighting. Or a speech-to-text system might not be used reliably to provide closed captions for online lectures because it fails to handle technical jargon.
- The authors should discuss the computational efficiency of the proposed algorithms and how they scale with dataset size.
- If applicable, the authors should discuss possible limitations of their approach to address problems of privacy and fairness.
- While the authors might fear that complete honesty about limitations might be used by reviewers as grounds for rejection, a worse outcome might be that reviewers discover limitations that aren't acknowledged in the paper. The authors should use their best judgment and recognize that individual actions in favor of transparency play an important role in developing norms that preserve the integrity of the community. Reviewers will be specifically instructed to not penalize honesty concerning limitations.

3. **Theory Assumptions and Proofs**

Question: For each theoretical result, does the paper provide the full set of assumptions and a complete (and correct) proof?

Answer: [Yes]

Justification: In the main text and the Appendix.

Guidelines:

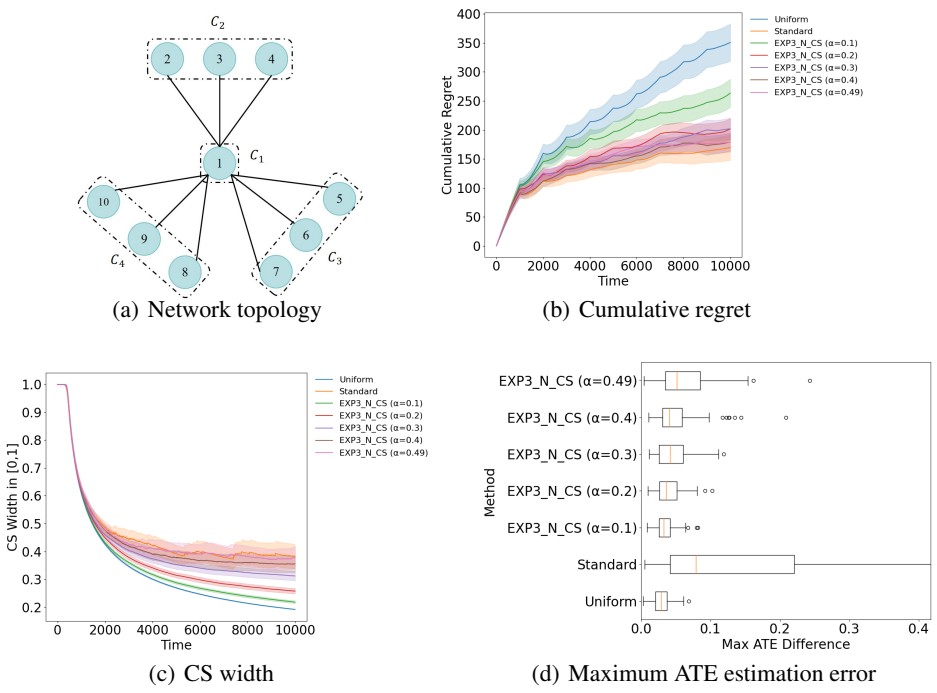

(a) Network topology

(b) Cumulative regret

(c) CS width

(d) Maximum ATE estimation error

Figure 6: Experimental results of instance 4.

- The answer NA means that the paper does not include theoretical results.
- All the theorems, formulas, and proofs in the paper should be numbered and cross-referenced.
- All assumptions should be clearly stated or referenced in the statement of any theorems.
- The proofs can either appear in the main paper or the supplemental material, but if they appear in the supplemental material, the authors are encouraged to provide a short proof sketch to provide intuition.
- Inversely, any informal proof provided in the core of the paper should be complemented by formal proofs provided in appendix or supplemental material.
- Theorems and Lemmas that the proof relies upon should be properly referenced.

4. **Experimental Result Reproducibility**

Question: Does the paper fully disclose all the information needed to reproduce the main experimental results of the paper to the extent that it affects the main claims and/or conclusions of the paper (regardless of whether the code and data are provided or not)?

Answer: [Yes]

Justification: In Section 6 and the Appendix.

Guidelines:

- The answer NA means that the paper does not include experiments.
- If the paper includes experiments, a No answer to this question will not be perceived well by the reviewers: Making the paper reproducible is important, regardless of whether the code and data are provided or not.
- If the contribution is a dataset and/or model, the authors should describe the steps taken to make their results reproducible or verifiable.
- Depending on the contribution, reproducibility can be accomplished in various ways. For example, if the contribution is a novel architecture, describing the architecture fully might suffice, or if the contribution is a specific model and empirical evaluation, it may be necessary to either make it possible for others to replicate the model with the same dataset, or provide access to the model. In general, releasing code and data is often

one good way to accomplish this, but reproducibility can also be provided via detailed instructions for how to replicate the results, access to a hosted model (e.g., in the case of a large language model), releasing of a model checkpoint, or other means that are appropriate to the research performed.

- While NeurIPS does not require releasing code, the conference does require all submissions to provide some reasonable avenue for reproducibility, which may depend on the nature of the contribution. For example
  (a) If the contribution is primarily a new algorithm, the paper should make it clear how to reproduce that algorithm.
  (b) If the contribution is primarily a new model architecture, the paper should describe the architecture clearly and fully.
  (c) If the contribution is a new model (e.g., a large language model), then there should either be a way to access this model for reproducing the results or a way to reproduce the model (e.g., with an open-source dataset or instructions for how to construct the dataset).
  (d) We recognize that reproducibility may be tricky in some cases, in which case authors are welcome to describe the particular way they provide for reproducibility. In the case of closed-source models, it may be that access to the model is limited in some way (e.g., to registered users), but it should be possible for other researchers to have some path to reproducing or verifying the results.

5. **Open access to data and code**

Question: Does the paper provide open access to the data and code, with sufficient instructions to faithfully reproduce the main experimental results, as described in supplemental material?

Answer: [Yes]

Justification: In Section 6.

Guidelines:

- The answer NA means that paper does not include experiments requiring code.
- Please see the NeurIPS code and data submission guidelines (https://nips.cc/public/guides/CodeSubmissionPolicy) for more details.
- While we encourage the release of code and data, we understand that this might not be possible, so "No" is an acceptable answer. Papers cannot be rejected simply for not including code, unless this is central to the contribution (e.g., for a new open-source benchmark).
- The instructions should contain the exact command and environment needed to run to reproduce the results. See the NeurIPS code and data submission guidelines (https://nips.cc/public/guides/CodeSubmissionPolicy) for more details.
- The authors should provide instructions on data access and preparation, including how to access the raw data, preprocessed data, intermediate data, and generated data, etc.
- The authors should provide scripts to reproduce all experimental results for the new proposed method and baselines. If only a subset of experiments are reproducible, they should state which ones are omitted from the script and why.
- At submission time, to preserve anonymity, the authors should release anonymized versions (if applicable).
- Providing as much information as possible in supplemental material (appended to the paper) is recommended, but including URLs to data and code is permitted.

6. **Experimental Setting/Details**

Question: Does the paper specify all the training and test details (e.g., data splits, hyperparameters, how they were chosen, type of optimizer, etc.) necessary to understand the results?

Answer: [Yes]

Justification: In Section 6 and the Appendix.

Guidelines:

- The answer NA means that the paper does not include experiments.
- The experimental setting should be presented in the core of the paper to a level of detail that is necessary to appreciate the results and make sense of them.
- The full details can be provided either with the code, in appendix, or as supplemental material.

7. **Experiment Statistical Significance**

   Question: Does the paper report error bars suitably and correctly defined or other appropriate information about the statistical significance of the experiments?

   Answer: [No]

   Justification: Error bars are not reported because it would be too computationally expensive.

   Guidelines:

   - The answer NA means that the paper does not include experiments.
   - The authors should answer "Yes" if the results are accompanied by error bars, confidence intervals, or statistical significance tests, at least for the experiments that support the main claims of the paper.
   - The factors of variability that the error bars are capturing should be clearly stated (for example, train/test split, initialization, random drawing of some parameter, or overall run with given experimental conditions).
   - The method for calculating the error bars should be explained (closed form formula, call to a library function, bootstrap, etc.)
   - The assumptions made should be given (e.g., Normally distributed errors).
   - It should be clear whether the error bar is the standard deviation or the standard error of the mean.
   - It is OK to report 1-sigma error bars, but one should state it. The authors should preferably report a 2-sigma error bar than state that they have a 96% CI, if the hypothesis of Normality of errors is not verified.
   - For asymmetric distributions, the authors should be careful not to show in tables or figures symmetric error bars that would yield results that are out of range (e.g. negative error rates).
   - If error bars are reported in tables or plots, The authors should explain in the text how they were calculated and reference the corresponding figures or tables in the text.

8. **Experiments Compute Resources**

   Question: For each experiment, does the paper provide sufficient information on the computer resources (type of compute workers, memory, time of execution) needed to reproduce the experiments?

   Answer: [Yes]

   Justification: In Section 6 and the Appendix.

   Guidelines:

   - The answer NA means that the paper does not include experiments.
   - The paper should indicate the type of compute workers CPU or GPU, internal cluster, or cloud provider, including relevant memory and storage.
   - The paper should provide the amount of compute required for each of the individual experimental runs as well as estimate the total compute.
   - The paper should disclose whether the full research project required more compute than the experiments reported in the paper (e.g., preliminary or failed experiments that didn't make it into the paper).

9. **Code Of Ethics**

   Question: Does the research conducted in the paper conform, in every respect, with the NeurIPS Code of Ethics https://neurips.cc/public/EthicsGuidelines?

   Answer: [Yes]

   Justification: In Section 6.

Guidelines:

- The answer NA means that the authors have not reviewed the NeurIPS Code of Ethics.
- If the authors answer No, they should explain the special circumstances that require a deviation from the Code of Ethics.
- The authors should make sure to preserve anonymity (e.g., if there is a special consideration due to laws or regulations in their jurisdiction).

10. **Broader Impacts**

Question: Does the paper discuss both potential positive societal impacts and negative societal impacts of the work performed?

Answer: [NA]

Justification: There is no societal impact of the work performed.

Guidelines:

- The answer NA means that there is no societal impact of the work performed.
- If the authors answer NA or No, they should explain why their work has no societal impact or why the paper does not address societal impact.
- Examples of negative societal impacts include potential malicious or unintended uses (e.g., disinformation, generating fake profiles, surveillance), fairness considerations (e.g., deployment of technologies that could make decisions that unfairly impact specific groups), privacy considerations, and security considerations.
- The conference expects that many papers will be foundational research and not tied to particular applications, let alone deployments. However, if there is a direct path to any negative applications, the authors should point it out. For example, it is legitimate to point out that an improvement in the quality of generative models could be used to generate deepfakes for disinformation. On the other hand, it is not needed to point out that a generic algorithm for optimizing neural networks could enable people to train models that generate Deepfakes faster.
- The authors should consider possible harms that could arise when the technology is being used as intended and functioning correctly, harms that could arise when the technology is being used as intended but gives incorrect results, and harms following from (intentional or unintentional) misuse of the technology.
- If there are negative societal impacts, the authors could also discuss possible mitigation strategies (e.g., gated release of models, providing defenses in addition to attacks, mechanisms for monitoring misuse, mechanisms to monitor how a system learns from feedback over time, improving the efficiency and accessibility of ML).

11. **Safeguards**

Question: Does the paper describe safeguards that have been put in place for responsible release of data or models that have a high risk for misuse (e.g., pretrained language models, image generators, or scraped datasets)?

Answer: [NA]

Justification: The paper poses no such risks.

Guidelines:

- The answer NA means that the paper poses no such risks.
- Released models that have a high risk for misuse or dual-use should be released with necessary safeguards to allow for controlled use of the model, for example by requiring that users adhere to usage guidelines or restrictions to access the model or implementing safety filters.
- Datasets that have been scraped from the Internet could pose safety risks. The authors should describe how they avoided releasing unsafe images.
- We recognize that providing effective safeguards is challenging, and many papers do not require this, but we encourage authors to take this into account and make a best faith effort.

12. **Licenses for existing assets**

Question: Are the creators or original owners of assets (e.g., code, data, models), used in the paper, properly credited and are the license and terms of use explicitly mentioned and properly respected?

Answer: [Yes] .

Justification: In Section 6.

Guidelines:

- The answer NA means that the paper does not use existing assets.
- The authors should cite the original paper that produced the code package or dataset.
- The authors should state which version of the asset is used and, if possible, include a URL.
- The name of the license (e.g., CC-BY 4.0) should be included for each asset.
- For scraped data from a particular source (e.g., website), the copyright and terms of service of that source should be provided.
- If assets are released, the license, copyright information, and terms of use in the package should be provided. For popular datasets, `paperswithcode.com/datasets` has curated licenses for some datasets. Their licensing guide can help determine the license of a dataset.
- For existing datasets that are re-packaged, both the original license and the license of the derived asset (if it has changed) should be provided.
- If this information is not available online, the authors are encouraged to reach out to the asset's creators.

13. **New Assets**

Question: Are new assets introduced in the paper well documented and is the documentation provided alongside the assets?

Answer: [NA]

Justification: The paper does not release new assets

Guidelines:

- The answer NA means that the paper does not release new assets.
- Researchers should communicate the details of the dataset/code/model as part of their submissions via structured templates. This includes details about training, license, limitations, etc.
- The paper should discuss whether and how consent was obtained from people whose asset is used.
- At submission time, remember to anonymize your assets (if applicable). You can either create an anonymized URL or include an anonymized zip file.

14. **Crowdsourcing and Research with Human Subjects**

Question: For crowdsourcing experiments and research with human subjects, does the paper include the full text of instructions given to participants and screenshots, if applicable, as well as details about compensation (if any)?

Answer: [NA]

Justification: The paper does not involve crowdsourcing nor research with human subjects.

Guidelines:

- The answer NA means that the paper does not involve crowdsourcing nor research with human subjects.
- Including this information in the supplemental material is fine, but if the main contribution of the paper involves human subjects, then as much detail as possible should be included in the main paper.
- According to the NeurIPS Code of Ethics, workers involved in data collection, curation, or other labor should be paid at least the minimum wage in the country of the data collector.

15. **Institutional Review Board (IRB) Approvals or Equivalent for Research with Human Subjects**

Question: Does the paper describe potential risks incurred by study participants, whether such risks were disclosed to the subjects, and whether Institutional Review Board (IRB) approvals (or an equivalent approval/review based on the requirements of your country or institution) were obtained?

Answer: [NA]

Justification: The paper does not involve crowdsourcing nor research with human subjects

Guidelines:

- The answer NA means that the paper does not involve crowdsourcing nor research with human subjects.
- Depending on the country in which research is conducted, IRB approval (or equivalent) may be required for any human subjects research. If you obtained IRB approval, you should clearly state this in the paper.
- We recognize that the procedures for this may vary significantly between institutions and locations, and we expect authors to adhere to the NeurIPS Code of Ethics and the guidelines for their institution.
- For initial submissions, do not include any information that would break anonymity (if applicable), such as the institution conducting the review.

