# OpenReview forum: "Design-Based Bandits Under Network Interference: Trade-Off Between Regret and Statistical Inference"
_NeurIPS.cc/2025/Conference — NeurIPS 2025 poster_

### Official Review · Reviewer_LDZS · 2025-07-03

**Clarity:** 3
**Significance:** 3
**Originality:** 3
**Rating:** 5
**Confidence:** 3

**Summary:**

This paper studies an adversarial bandit problem with network interference where in each round, the decision maker must assign one of K arms to each of the N nodes in a network with some underlying graph structure. The authors study three (generally) competing objectives of regret minimization, validity of post hoc inference and average-treatment-effect estimation and establish Pareto optimality results. They also propose an anytime-valid asymptotic confidence sequence and based on it an EXP3-style algorithm that balances the trade-off between regret minimization and inference accuracy.

**Questions:**

When N=1, does Theorem 4.1 reduce to Theorem 1 of Simchi-Levi and Wang [2024]? Or Condition 3.1 prohibits this reduction? Further to this point, it would be nice to know how much of the proof of Theorem 4.1 is adapted from the proof of Theorem 1 of Simchi-Levi and Wang [2024].

In terms of the algorithm, Simchi-Levi and Wang [2024] also consider a variant of EXP3. Given this, it would be helpful for the readers to know what new technical/analytical contributions the authors make to overcome limitations of prior techniques.

**Ethical Concerns:**

["NO or VERY MINOR ethics concerns only"]

**Final Justification:**

The author response was comprehensive and helped me contextualize their results better. It also resolved my initial concern (favorably) about differentiation from prior work.

**Limitations:**

Yes

**Quality:**

3

**Strengths And Weaknesses:**

Strengths: The paper is well-written generally and makes progress on an important problem in causal learning. Addressing causal inference with interference in an online learning setting is quite relevant in practical applications like social networks and large-scale experimentation on platforms. The anytime-valid confidence sequences provided in this paper are important in this context. The authors also generalize the theoretical lower bound of (Simchi-Levi and Wang, 2024) to capture the complexity of the networked setting via a new implicit primitive (effective arm space size).

The following are not weaknesses, just some resistance I encountered while parsing the content:
Some constructs (e.g., exposure mappings and clustering) are not common in traditional bandit literature and might need a little more context and/or instantiation via examples for readers not entirely familiar with the causal inference/bandit learning interface. I understand the paper is constrained to be within 9 pages, but it would be useful if the model development in S3.1 and S3.2 is pre-padded with a little more context in this regard. While not a weakness per se of the paper, it is less than ideal that the main algorithm itself is pushed to supplementary pages together with a high-level discussion of what it does. This leaves S5.2 somewhat obscure on the first read.

I want to emphasize that I liked reading the paper but my conservative score is a consequence of being unable to place the methodological and analytical contributions in this work vis-a-vis prior art. In particular, though the networked model under interference is a significant generalization of the simple setting of (Simchi-Levi and Wang, 2024), the hardness of analyzing this setting is not very clear. I would be open to revising my scores pending a satisfactory rebuttal to this end.

---

> ### Author Rebuttal · Authors · 2025-07-28
>
> Thanks for your comments! We will address it point by point.
>
> >***Q1: The following are not weaknesses, just some resistance I encountered while parsing the content: Some constructs (e.g., exposure mappings and clustering) are not common in traditional bandit literature and might need a little more context and/or instantiation via examples...While not a weakness per se of the paper, it is less than ideal that the main algorithm itself is pushed to supplementary pages together with a high-level discussion of what it does. This leaves S5.2 somewhat obscure on the first read.***
> ---
> **A1:** Exposure mapping (and clustering) is a well-established concept in causal inference and experimental design. Our goal is to extend this idea to the online learning and bandit setting. Intuitively, exposure mapping embeds the exponentially large action space $K^N$ into a lower-dimensional space or manifold of size $|\mathcal{U}_{\mathcal{E}}|$, via a mapping defined as:
>
> $$
> s \equiv \mathbf{S}(i, A, \mathbb{H}), \quad \text{where } \mathbf{S}: \mathcal{U} \times \mathcal{K}^{\mathcal{U}} \times \mathbb{H} \rightarrow \mathcal{U}_s.
> $$
>
> For example, in real-world scenarios such as COVID-19 treatment or ride-sharing dispatch, the exposure mapping can encode the proportion of a unit’s neighbors who are treated (e.g., receiving medication or requesting a ride). Based on this, we further apply clustering to reduce the dimensionality of the exposure-based arm space, ensuring that the action space $|\mathcal{U}_{E}|$ remains computationally feasible. In the revised version, we will provide a more intuitive exposition of the exposure mapping and move the main algorithm into the main text to enhance readability.
>
>
>
>
> >***Q2: When N=1, does Theorem 4.1 reduce to Theorem 1 of Simchi-Levi and Wang (2024)? Or Condition 3.1 prohibits this reduction? Further to this point, it would be nice to know how much of the proof of Theorem 4.1 is adapted from the proof of Theorem 1 of Simchi-Levi and Wang [2024]. In particular, though the networked model under interference is a significant generalization of the simple setting of (Simchi-Levi and Wang, 2024), the hardness of analyzing this setting is not very clear. I would be open to revising my scores pending a satisfactory rebuttal to this end.***
> ---
> **A2:**
> When $N = 1$, our setting naturally reduces to a case similar to that of Simchi-Levi and Wang (2024). However, our results are **strictly stronger**: we not only retain tightness in the time horizon $T$, but also in the total action space dimension. This extension is both necessary and important—under network interference, the action space can grow exponentially large, rendering the conclusions in Simchi-Levi and Wang \[2024] inapplicable.
>
> For example, even if we take a simplified exposure mapping where each unit's exposure is defined by its cluster index (with $M$ clusters), the resulting action space already contains $2^{d_s}$ possibilities. In practice, our exposure arm space can be much richer—for instance, encoding the proportion of treated neighbors—which is essential for capturing a broad range of network interference scenarios.
>
> This conceptual advancement naturally gives rise to new technical challenges, beyond those addressed in Simchi-Levi and Wang (2024). We summarize these challenges into three key dimensions:
>
>
>
> **``Global Counterexample Construction under Network Interference``**
> In the Simchi-Levi et al., (2024), the lower bound is constructed via a simple reward gap between two arms—localized and independent across individuals. In contrast, our setting involves network interference, where the reward function depends on the joint action profile across the network. This requires designing globally distinguishable yet locally indistinguishable problem instances—i.e., constructing interference-aware reward functions that are information-theoretically hard to distinguish under a limited number of observations. This significantly increases the complexity of the lower bound argument.
>
> **``Robustness to Misspecification and Approximation Error``**
> In our case, we could further explicitly account for misspecification between the inference mechanism (e.g., IPW/REG/DR estimators) and the actual exposure structure or clustering granularity. As a result, our analysis must control for functional approximation error in the estimation layer, which creates an additional regret–inference tradeoff not addressed in prior work. We give a brief illustration: Let $\hat\Delta_T(S_i,S_j)$  be our IPW estimator based on the *designed* exposure super-arms $S_i,S_j\in\mathcal U_{E}$. Denote by $\bar{\tau}^{\star}_T(S_i, S_j)$ the *true* ATE computed under the (possibly finer) ground-truth exposure mapping $S^{\star}$. Define the ``approximation (misspecification) bias`` $\mathcal B(\mathbf S,\mathcal C)$ as follows:
>
> $\max_{S_i,S_j\in\mathcal U_{\mathcal{E}}} := |\bar{\tau}^{\star}_T(S_i,S_j)-\bar{\tau}_T(S_i,S_j)|,$ we prove the *biased* minimax lower bound
>
> $\min_{\pi, \hat{\Delta}} \max_{v \in \mathcal{E}_0}\sqrt{R(T, \pi)} \cdot (e(T, \hat{\Delta}) - \mathcal B(\mathbf S,\mathcal C) )= {\Omega}(\sqrt{|\mathcal{U}_E|}).$
>
>
> Intuitively, misspecification bias introduces larger errors. However, we can maintain robustness by controlling the term $(e(T, \hat{\Delta}) - \mathcal B(\mathbf S,\mathcal C) $ to preserve our lower bound to show robustness: ``even after subtracting the systematic bias, the regret–inference product cannot beat the lower bound barrier``; it is strictly higher than in the perfectly-specified setting in the previous literature. We have added it in our revised version.
>
>
> **``Transformation between Adversarial & Stochastic Settings``**
> Simchi(2024) operates entirely in a stochastic regime. Our analysis, by contrast, develops the lower bound in an adversarial design-based framework, and further shows that this bound recovers the classical stochastic lower bound under special cases of the exposure mapping. This unification reveals that both regimes share a common fundamental tradeoff structure and establishes a stronger minimax lower bound valid under broader assumptions.
>
>
>
>
>
> >***Q3: In terms of the algorithm, Simchi-Levi and Wang (2024) also consider a variant of EXP3. Given this, it would be helpful for the readers to know what new technical/analytical contributions the authors make to overcome limitations of prior techniques.***
> ---
> **A3:** Thank you for the suggestion. We have now clarified the technical/conceptual contributions of our algorithm.
>
> 1. We develop a uniform analysis framework that enables our MAD design, combined with an asymptotic confidence sequence, to be seamlessly integrated into any general regret minimization algorithm in adversarial settings. Note that the MAD strategy takes the form $\pi_t^{\text{MAD}}(S) = \frac{1}{|U_E|} \delta_t + (1 - \delta_t) \pi_t^{\text{ALG}}(S)$, where the first term (with coefficient $\delta_t$) is introduced to improve ATE estimation, and the second term $\pi_t^{\text{ALG}}(S)$ is the base algorithm aimed at regret minimization. To analyze the regret, we decompose the total regret into two components corresponding to the two terms in the strategy, and then separately upper bound each. As shown in Theorem 5.6 (ii), the first part of the regret scales as $\sqrt{|U_{E}| T}$, which comes from the $(1 - \delta_t)\pi_t^{\text{ALG}}(S)$ term (in our case, a network variant of EXP3), and the second part scales as $T^{1 - \alpha}$, arising from the $\frac{1}{|U_{E}|} \delta_t$ term, which is independent of the specific base algorithm. Therefore, we can easily analyze the overall regret upper bound as $\tilde{O}(X + T^{1 - \alpha})$, where $X$ denotes the regret of the base algorithm. In contrast, the method by Simchi-Levi and Wang (2024) improves ATE estimation by adding extra exploration that depends heavily on the structure of his elimination-based EXP3.  Their tight coupling makes it non-trivial to extend such sampling strategies to other adversarial bandit algorithms, such as EXP3-IX.
>
> 2. Simchi-Levi and Wang (2024)'s algorithm is heavily depended on the arm number $K$, i.e., $R(T, \pi) = \tilde{O}(K^5 + T^{1 - \alpha}) \quad (\alpha \in [0, 1])$ and $e(T, \hat{\Delta}) = \tilde{O}(K^2 T^{\alpha/2 - 1/2})$. This result is largely due to the inefficiency of their additional exploration strategy. Moreover, their theoretical guarantees rely on the assumption that the reward gap between the optimal and suboptimal arms is $\Theta(1)$, which limits generality. While this may be acceptable in Chonghuan’s networkless setting, where $K$ can be treated as a relatively insignificant constant, our exposure action set $U_E$ intertwines $K$ with other critical quantities—such as the number of nodes $N$ and the degree of interference—thereby making it a central parameter in our analysis. As a result, directly extending Simchi-Levi and Wang (2024)’s method to the networked setting is less effective. By leveraging the modularity of MAD, our framework can easily decouple additional exploration from regret minimization. As a result, we obtain regret and ATE bounds $R(T, \pi) = \tilde{O}(\sqrt{|U_{E}|T} + T^{1 - \alpha})$ for $\alpha \in [0, 1/4]$, and $e(T, \hat{\Delta}) = \tilde{O}(|U_E| T^{\alpha - 1/2})$, which are significantly less sensitive to the size of $U_E$.
>
> **Reference:**
>
> Simchi-Levi, David and Wang, Chonghuan, Multi-armed Bandit Experimental Design: Online Decision-making and Adaptive Inference (September 20, 2022). Available at SSRN.
>
> Zhang Z, Wang Z. Online experimental design with estimation-regret trade-off under network interference[J]. arXiv preprint arXiv:2412.03727, 2024.
>
> Michael P. Leung, Causal Inference Under Approximate Neighborhood Interference.

---

> > ### Comment · Reviewer_LDZS · 2025-08-06
> >
> > I thank the authors for offering a clear explanation. I would like the revision to reflect some of these points so the contributions can be seen more clearly. I am revising my rating and vote for acceptance.

---

> ### Author Response · Authors · 2025-08-06
> **Thank you for your positive feedback**
>
> We sincerely thank you for your thoughtful response. We will carefully revise the manuscript to ensure the key contributions are better highlighted, as you suggested.

---

### Official Review · Reviewer_8D4t · 2025-07-03

**Clarity:** 3
**Significance:** 3
**Originality:** 3
**Rating:** 5
**Confidence:** 3

**Summary:**

Briefly, the paper presents a modular framework for Multi-Armed Bandits with Network Interference (MABNI). It contributes the trade-off between regret minimization and inference accuracy through establishing a theoretical Pareto frontier for minimizing regret and ATE estimation error in adversarial networked bandit scenarios and also their EXP3 algorithm to address regret minimization, continual inference and ATE estimation error minimization. They manage the action space with exposure mapping and clustering.

**Questions:**

How is "overly prioritizing the optimal arm can compromise the inference accuracy of the sub-optimal arm" neglected? Isn't this a version of exploration-exploitation trade-off, which is pretty standard in bandit research?

The phrases "anytime-valid" and asymptotic do not seem complementary. Please elaborate.

In (1), how is the adjacency matrix H both a possible argument for the mapping S and part of the domain bold italic S?

Line 159: isn't the size reduced to a power of the size of $d_s$?

There seems to be mix-up with the notation in the first part of (2). Please properly define $S_t$.

Definition 3.2: explain ATE here.

(3): what is the usage of the subscript in R? Adversarial nature? It is undefined.

Also, why is the lower-bound in Theorem 3.1 of importance? Why is the multiplication of square-root regret and ATE error is specifically chosen for lower bounding?

The experiments section can be improved. The figures only support the theoretical findings in a basic manner. Maybe replace the figures with tables to properly report various experimental results.

**Ethical Concerns:**

["NO or VERY MINOR ethics concerns only"]

**Final Justification:**

My questions (and also many other concerns raised by the reviewers) were mostly answered. The authors improved the clarity through resolving ambiguities, better conveyed the significance via adequate explanations, and elevated the experimental evaluation to highlight the impact of their work.

**Limitations:**

no, you can include some comments in conclusion.

**Quality:**

3

**Strengths And Weaknesses:**

### Quality:
The submission is somewhat technically sound.
Claims are supported by theoretical analysis and experimental results.
The methods used are appropriate, each addressing different aspects in a modular fashion.
This is a complete piece of work, with some open problems as a future direction.
The authors are careful and honest about evaluating both the strengths and weaknesses of their work, but experiments section is a bit neglected.

### Clarity:
The submission is mostly clearly written.
It is generally well organized.
It seems to adequately inform the reader by providing enough information to reproduce its results.

### Significance:
The results are impactful for the community, with a well-establish trade-off between regret minimization and inference accuracy for consideration in critical practical applications.
Others (researchers or practitioners) are likely to use the ideas or build on them.
The submission addresses a difficult task in a better way than previous work, as demonstrated by the shortcomings of the past works.
It advances our understanding and knowledge on the topic in a demonstrable way. However, the importance of lower-bound is ambiguous and there exists an optimality gap.
It provides a unique theoretical modular framework.

### Originality:
The work provides new insights and deepen understanding for MABNI trade-off with pareto frontier.
It is clear how this work differs from previous contributions, with relevant citations provided.
The work introduces novel methods that advance the field.
This work offers a novel combination of existing techniques to address different objectives, and the reasoning behind this combination is well-articulated.

---

> ### Author Rebuttal · Authors · 2025-07-28
>
> Thank you for your thoughtful and encouraging evaluation. We’re glad that you found our work to be clearly written and technically sound. We have carefully addressed your concerns in our responses below.
>
> > ***Q1: How is "overly prioritizing the optimal arm can compromise the inference accuracy of the sub-optimal arm" neglected? Isn't this a version of exploration-exploitation trade-off, which is pretty standard in bandit research?***
>
> ---
>
> **A1:** Thank you for the insightful question. While this statement shares similarities with the standard exploration-exploitation trade-off in bandit problems, there are some important differences. In classical bandit problems focused solely on regret minimization, exploration is primarily used to (implicitly) identify the best arm. Once identified, the algorithm simply pulls it, and the learning objective is fulfilled. However, in our setting, even if the best arm is known, the task is not complete. We still need to allocate pulls to all suboptimal arms in order to accurately estimate their expected rewards and, consequently, the ATE. This introduces a more delicate trade-off, where overly prioritizing the best arm can compromise the accuracy of ATE estimation.
>
> > ***Q2: The phrases "anytime-valid" and asymptotic do not seem complementary. Please elaborate.***
>
> ---
>
> **A2:** Thank you for your suggestion. A confidence sequence (CS) $(I_t)_{t \ge 1}$ is called anytime-valid if: $\mathbb{P} (\forall t \geq 1,\ \theta^*_t \in I_t) \geq 1 - \delta$. This property ensures that the interval is valid at all time points (i.e., $t\ge1$) simultaneously.
>
> The notion of an asymptotic CS was first introduced by Waudby-Smith et al. (2021). Formally, define $\hat \mu_t \pm C_t^*$ as a non-asymptotic $(1 - \tilde \delta)$ CS for a sequence of target parameters $\mu_t$,
>
> and another CS $\hat \mu_t \pm \hat C_t$ such that $\hat C_t/C_t^* \xrightarrow{\text{a.s.}} 1$, then $\hat \mu_t \pm \hat C_t$ is said to be an asymptotic $(1 - \tilde{\delta})$ CS for $\mu_t$. A non-asymptotic confidence sequence with confidence level $1 - \tilde{\delta}$ is a sequence of intervals $\hat \mu_t \pm C_t^*$ such that, with probability at least $1 - \tilde{\delta}$, all intervals simultaneously cover the target parameters with probability at least $1 - \tilde{\delta}$.
>
> >***Q3: In (1), how is the adjacency matrix H both a possible argument for the mapping S and part of the domain bold italic S?***
> ---
> **A3:** Thank you for your careful reading. We acknowledge the confusion around the role of the adjacency matrix $H$ in the definition of the exposure mapping $S(\cdot)$. To clarify: in our framework, $H$ is treated as an **external input** to the exposure mapping $S$, i.e., $S_i = S(H, A)$, where $A$ is the treatment assignment. The function $S$ maps the global treatment and network context into a lower-dimensional summary (e.g., degree-weighted exposure). The bold italic $\mathcal{S}$ denotes the **codomain** or the **range** of possible exposure values, and does not itself contain $H$. To avoid ambiguity, we have revised the notation in the paper to make this distinction explicit: $S : \{0,1\}^N \times \mathbb{R}^{N \times N} \to \mathcal{S}$, where $H \in \mathbb{R}^{N \times N}$ is the adjacency matrix. We thank the reviewer for pointing this out.
>
>
>
>
> > ***Q4: Line 159: isn't the size reduced to a power of the size of $d_s$?***
>
> ---
>
> **A4:** Thank you for the suggestion. Yes, due to the exposure mapping and clustering procedures, the size of the exposure super arm set is reduced to $d_s^C$, where $d_s$ denotes the number of exposure arms and $C$ is the number of clusters. We have clarified this point in Line~178.
>
> > ***Q5: There seems to be mix-up with the notation in the first part of (2). Please properly define $S_t$.***
>
> ---
>
> **A5:** Thank you for your suggestion. In our paper, $S_t$ denotes the exposure super arm selected by the learning algorithm in round $t$.
>
>
>
> >***Q6: Definition 3.2: explain ATE***
> ---
> **A6:** The ATE (Average Treatment Effect) defined here intuitively captures: the average difference in group-level responses when assigning the entire population to two different exposure super arms $S_i$ and $S_j$ over multiple rounds. Formally,  it measures the mean outcome difference across $t$ rounds between these two exposure conditions.
>
>
>
> > ***Q7: What is the usage of the subscript in $R$? Adversarial nature? It is undefined.***
>
> ---
>
> **A7:** Thank you for your question. In Lines 139--141, we define $\mathcal E_0$ as the set of all legitimate instances of the adversarial bandit with network interference. The notation $\nu$ in our paper refers to a specific instance in $\mathcal E_0$. Therefore, in Eq.~(3), $R_{\nu}(\pi, T)$ denotes the regret incurred by algorithm $\pi$ under the adversarial instance $\nu \in \mathcal{E}_0$.
>
> > ***Q8: Why is the lower-bound in Theorem 3.1 of importance? Why is the multiplication of square-root regret and ATE error is specifically chosen for lower bounding?***
> ---
> **A8:** This is the standard paradigm to express the trade-off (it has been proved to be optimal in the stochastics setting), referring to Zhang et al and Simchi et al.
>
>
> > ***Q9: The experiments section can be improved. The figures only support the theoretical findings in a basic manner. Maybe replace the figures with tables to properly report various experimental results.***
>
> ---
>
> **A9:** Thank you for the helpful suggestion. In the revised manuscript, we have replaced the original figures with tables that more systematically summarize the experimental results. These tables report key metrics — including cumulative regret, confidence sequence width, and ATE estimation error — under different choices of the parameter $\alpha$. Following the experimental setup described in the main text, an example of the revised result tables is shown below:
>
> **Final Cumulative Regret (T = 10,000)**
>
>
> | Method      | Parameters    | Mean     | SE     |
> |-------------|--------------|----------|--------|
> | Uniform     | -            | 189.30   | 2.84   |
> | EXP3_N_CS   | $\alpha$=0.05| 167.25   | 2.36   |
> | EXP3_N_CS   | $\alpha$=0.1 | 152.97   | 2.14   |
> | EXP3_N_CS   | $\alpha$=0.15 | 143.78   | 2.05   |
> | EXP3_N_CS   | $\alpha$=0.2 | 137.99   | 1.99   |
> | EXP3_N_CS   | $\alpha$=0.24| 134.21   | 2.01   |
> | Standard    | -            | 127.09   | 2.01   |
>
>
>
> **Final Confidence Sequence Width (T = 10,000)**
>
> | Method      | Parameters    | Mean     | SE      |
> |-------------|--------------|----------|---------|
> | Uniform     | -            | 0.1924   | 0.0001  |
> | EXP3_N_CS   | $\alpha$=0.05| 0.1972   | 0.0003  |
> | EXP3_N_CS   | $\alpha$=0.1 | 0.2062   | 0.0006  |
> | EXP3_N_CS   | $\alpha$=0.15 | 0.2157 | 0.0009  |
> | EXP3_N_CS   | $\alpha$=0.2 | 0.2247   | 0.0012  |
> | EXP3_N_CS   | $\alpha$=0.24| 0.2362   | 0.0015  |
> | Standard    | -            | 0.2801   | 0.0042  |
>
>
>
> **ATE Difference Statistics (T = 10,000)**
>
>
> | Method      | Parameters    | Mean     | Median  | Std     |
> |-------------|--------------|----------|---------|---------|
> | Uniform     | -            | 0.0287   | 0.0274  | 0.0122  |
> | EXP3_N_CS   | $\alpha$=0.05| 0.0294   | 0.0285  | 0.0120  |
> | EXP3_N_CS   | $\alpha$=0.1 | 0.0307   | 0.0294  | 0.0131  |
> | EXP3_N_CS   | $\alpha$=0.15 | 0.0326   | 0.0303  | 0.0148  |
> | EXP3_N_CS   | $\alpha$=0.2 | 0.0334   | 0.0315  | 0.0152  |
> | EXP3_N_CS   | $\alpha$=0.24| 0.0364   | 0.0339  | 0.0172  |
> | Standard    | -            | 0.0658   | 0.0332  | 0.0882  |
>
>
> As shown in the tables, increasing $\alpha$ reduces cumulative regret but slightly increases the confidence sequence width and ATE estimation error. Notably, $\alpha = 0.24$ achieves a balanced trade-off: its regret is close to that of the Standard baseline (EXP3 with network), while its confidence sequence width and ATE estimation error remain close to those of the Uniform baseline (uniform exploration).
>
> **Reference:**
>
> Waudby-Smith I, Arbour D, Sinha R, et al. Time-uniform central limit theory and asymptotic confidence sequences[J]. The Annals of Statistics, 2024, 52(6): 2613-2640.
>
> Simchi-Levi, David and Wang, Chonghuan, Multi-armed Bandit Experimental Design: Online Decision-making and Adaptive Inference (September 20, 2022). Available at SSRN.
>
> Zhang Z, Wang Z. Online experimental design with estimation-regret trade-off under network interference[J]. arXiv preprint arXiv:2412.03727, 2024.

---

> > ### Comment · Reviewer_8D4t · 2025-08-06
> >
> > Thank you for the detailed response. I have read and considered it. It mostly addresses my concerns. I maintain my favorable impression towards acceptance.

---

> ### Author Response · Authors · 2025-08-06
> **Thank you for your positive feedback**
>
> We appreciate your careful consideration of our response and are glad it was helpful in addressing your concerns.

---

### Official Review · Reviewer_fNDP · 2025-07-03

**Clarity:** 3
**Significance:** 3
**Originality:** 3
**Rating:** 5
**Confidence:** 3

**Summary:**

The paper studies the multi-armed bandit problem with network interference problem. The goal is to design an algorithm that continually satisfies two-fold objectives: regret minimisation, and accurate ATE estimation. The paper proposes first a lower bound on the product of regret and ATE estimation error. Then under known adjacency matrix assumption, it develops an adversarial bandit algorithm EXP3-N-CS, and proves the regret and ATE estimation upper bounds for it. Numerical experiments are provided to demonstrate efficiency of EXP3-N-CS.

**Questions:**

1. Check the weaknesses and please address them.
2. Why in theorem 5.5. $\alpha$ has to be in the range of $[0,1/4)$? What is the technical bottleneck or is it some step in proof that leads to it?
3. Can you provide a detailed ablation study on choice of $\alpha$?
4. Also, it would be interesting to see how the proposed algorithm performs against Simchi-Levi and Wang [2024] and Liang and Bojinov [2023] in network-less setup and Zhang and Wang [2024] in the networked setup. Specially, I am curious why to choose adversarial or stochastic algorithm in this setting.

**Ethical Concerns:**

["NO or VERY MINOR ethics concerns only"]

**Final Justification:**

The authors addressed all the questions thoroughly and promised to incorporate the changes. After reading other reviews and responses, I feel that together with the recommended addendum, it is an interesting work.

**Limitations:**

Except the known H assumption, limitations are not discussed much. Check pointers regarding $\alpha$ and tightness claims.

**Paper Formatting Concerns:**

N.A.

**Quality:**

3

**Strengths And Weaknesses:**

Strengths:
1. The paper is cleanly and logically written. Works as a good introduction to the setup for the reader.
2. The idea of deriving a "Pareto optimality" lower bound and upper bound in the setting as in [Simchi-Levi and Wang, 23] is interesting.
3. The experimental evidence across networks validates the applicability of the algorithm.

Weakness:
1. Please add the algorithm pseudocode in the main paper. It is very hard to understand Sec 5.2. without going to and fro to the Appendix.
2. It is not clear how to choose $\alpha$ in practice.
3. I understand why knowing H is helpful here and not knowing it would increase complexity. But I am not sure that it has to be known. The exposure mapping can be learned on-the-go too.
4.  The paper does not show why the produce of two objectives is "the" way to look into Pareto optimality in this setting. It is not intuitive, in general. An explanation is needed.
5. Line 362-365 were not necessary. You do not have to be slightly suboptimal or really optimal to write a science paper. Do remove this claim. Rather, Theorem 5.6. says that the gap between lower bound and upper bound grows with $T^{\alpha/2}$. It is more useful to community if you explain from where this gap is arising. Is it a proof issue or natural to the algorithm? Also, experiments do not point to "optimality". There is no lower bound plot in experiments.

---

> ### Author Rebuttal · Authors · 2025-07-28
>
> Thank you for your positive feedback. We're glad that you found the paper clearly written, the Pareto-optimality formulation interesting, and the experimental results convincing. We have carefully addressed your comments and suggestions in our responses below.
>
> > ***Q1: Please add the algorithm pseudocode in the main paper.***
>
> ---
>
> **A1:** Thanks for the suggestion. We will include the pseudocode in Section 5 in the future version of the paper.
>
> > ***Q2: It is not clear how to choose α in practice.***
>
> ---
>
> **A2:** Thank you for the question. A concrete example arises in evaluating treatment strategies for infectious disease. When facing a group of critically ill patients, it is important to prioritize regret minimization by assigning treatments that are currently believed to be most effective, thereby maximizing the overall chance of survival within the network. In contrast, for a group of patients with milder symptoms and stable conditions, assigning less-certain but promising treatments can facilitate a better evaluation of the relative effectiveness of different options. This leads to more accurate ATE estimation and supports improved treatment decisions for future populations. Accordingly, one may prefer choosing $\alpha$ closer to $1/4$ in the former case and closer to $0$ in the latter. We will include this example in a future version of the paper.
>
> >***Q3: I understand why knowing H is helpful here and not knowing it would increase complexity. But I am not sure that it has to be known.***
> ---
> **A3:** We emphasize that whether the adjacency matrix **$H$** must be known a priori depends entirely on how the exposure mapping $S(\cdot)$ is defined. For example, if our setting reduces to the scenario in Agarwal et al. (2024)—namely, when the exposure mapping depends on **all first-order neighbours**—then the neighbourhood information in $H$ must indeed be known in advance. In contrast, if our exposure mapping simply uses each node’s cluster index, then $H$ can remain unknown. Overall, we include $H$ in our setup in order to focus on a unified framework, and this does not imply that all information in $H$ always needs to be learned. Furthermore, thanks to our framework’s extensibility, the exposure mapping itself **can be learned and optimized**—we could show a combined approach with Shishir Adhikari et al. (2025) in the revised version.
>
>
> >**Q4: Why the product of two objectives is "the" way to look into Pareto optimality in this setting**
>
> **A4:** It is because the optimal solution (find the best partial order) of the mini-max problem $min_{\pi, \Delta} max_{\nu ∈ \mathcal E_0} \\{\sqrt{R_\nu(T, \pi)}, e_\nu(T, \Delta) \\}$ induce the form of production. We have added this formulation to the revised version.
>
>
>
>
>
>
>
> > ***Q5:  Please remove the claim in Lines 362–365 and clarify the reason of the gap between the lower and upper bounds — is it due to a proof issue or inherent to the algorithm? Additionally, justify why Theorem 5.5 restricts $\alpha \in [0, 1/4)$. Finally, the experiments do not demonstrate "optimality"; a lower bound curve is missing.***
>
> ---
>
> **A5:** Thank you for the insightful suggestion. We will remove the claim given in Lines 362–365.
>
> We believe the gap between the upper and lower bounds can be attributed to inherent structural constraints required by asymptotic confidence sequences (CS). Proposition 2.5 together with Condition L3 of Waudby--Smith et al. (2021) gives a convenient way to construct a valid $(1-\tilde{\delta})$ asymptotic CS in our learning setup: Define
> $\tilde V_t(S_i, S_j)
> = \sum_{s=1}^t \sigma_{s}^2(S_i, S_j)
> = \sum_{s=1}^t \left(\frac{\mathbb{E}[R_s(S_i)^2]}{\pi^{\text{MAD}}s(S_i)}+\frac{ \mathbb{E}[R_s(S_i)^2]}{\pi^{\text{MAD}}s(S_j) }\right)$.
> If there exists $\hat{V}_t(S_i,S_j)$ such that $t^{-1}(\hat V_t-\tilde V_t)(S_i,S_j) \xrightarrow{a.s.}0$, then $\hat{\bar{\tau}}_t(S_i, S_j) \pm \hat C_t(S_i, S_j)$ is a valid ($1-\tilde{\delta}$) asymptotic CS, where $\hat C_t(S_i,S_j)=\sqrt{\frac{2(\hat V_t(S_i,S_j)\eta^2+1)}{t^2\eta^2}
> \log\left(\frac{\sqrt{\hat V_t(S_i,S_j)\eta^2+1}}{\tilde{\delta}}\right)}$.
>
> To ensure $t^{-1}(\hat V_t-\tilde V_t)(S_i,S_j) \xrightarrow{a.s.}0$, we construct $\hat V_t(S_i,S_j)$ via the unbiased IPW estimator that is fully computable from the data: $\hat V_t(S_i, S_j) = \sum_{s=1}^t  \frac{ \mathbf{1}( S_{s} = S_i ) \cdot R_{s}(S_i)^2 }{ ( \pi^{\text{MAD}}s(S_i) )^2 }  + \frac{ \mathbf{1}( S_{s} = S_j ) \cdot R_{s}(S_j)^2 }{ ( \pi^{\text{MAD}}s(S_j) )^2 }$. Let $M_s(S_i,S_j):=(\hat \sigma_s^2- \sigma_s^2)(S_i,S_j)$. Then $M_s(S_i,S_j)$ is a martingale difference sequence and $t^{-1}\sum_{s=1}^t M_s(S_i,S_j)= t^{-1}(\hat V_t-\tilde V_t)(S_i,S_j)$. By the SLLN for martingale difference sequence, if $\sum_{t=1}^\infty \mathbb{E}[M_t^2(S_i,S_j)]/t^2<\infty$, we obtain $t^{-1}(\hat V_t-\tilde V_t)(S_i,S_j)\xrightarrow{a.s.}0$. This requires $(\hat \sigma_t^2 - \sigma_t^2)^2(S_i,S_j) = o(t)$ (by a $p$-series test).
> In our design this summability is guaranteed by $1/\pi_t^{\mathrm{MAD}}(S)=o(t^{1/4})$ for all $S\in \mathcal{U}_E$. Hence, when $\pi_t^{\mathrm{MAD}}(S)\propto t^{-\alpha}$, we require $\alpha\in[0,1/4)$. Under this requirement the CS width behaves as $\hat C_t=\tilde{O}(t^{\alpha-1/2})\quad \text{for }\alpha\in[0,1/4)$,
> and enforcing $1/\pi_t^{\mathrm{MAD}}(S)=o(t^{\alpha})$ for all $S\in\mathcal{U}_E$ induces additional regret $\tilde{O}(t^{1-\alpha})$. This ultimately leads to the gap between the lower and upper bounds.
>
> Following your suggestion, we will incorporate the above discussion into the main text for clarity. We will also add the corresponding lower bound to the experimental plots.
>
> > ***Q6: Can you provide a detailed ablation study on choice of $\alpha$?***
>
> ---
>
> **A6:** Thank you for your suggestion. We have included an ablation study evaluating the performance of our method under various choices of $\alpha$. Please see the tables and discussion provided in A8 of our rebuttal to Reviewer 8D4t, we apologize for this due to space constraints.
>
> > ***Q7: Also, it would be interesting to see how the proposed algorithm performs against Simchi-Levi and Wang [2024] and Liang and Bojinov [2023] in network-less setup and Zhang and Wang [2024] in the networked setup.***
>
> ---
>
> **A7:** Thank you for your suggestion. We have included a comparison here.
>
> In the stochastic setting without a network, the algorithm proposed by Simchi-Levi and Wang (2024) achieves a regret upper bound of $R(T, \pi) = \tilde{O}(K^5 + T^{1-\alpha}) \quad (\alpha \in [0,1])$, which relies on an additional assumption that the reward gap between the optimal and suboptimal arms is $\Theta(1)$, and an ATE estimation error upper bound $e(T, \hat{\Delta}) = \tilde{O}(K^2 T^{\alpha/2 - 1/2})$, leading to $\sqrt{R(T, \pi)} \cdot e(T, \hat{\Delta}) = \tilde{O}(K^2)$. In comparison, our algorithm achieves $R(T, \pi) = \tilde{O}(\sqrt{KT} + T^{1-\alpha})$ $(\alpha \in [0,1/4])$, $e(T, \hat{\Delta}) = \tilde{O}(KT^{\alpha - 1/2})$, $\sqrt{R(T, \pi)} \cdot e(T, \hat{\Delta}) = \tilde{O}(KT^{\alpha/2})$. Thus, our algorithm significantly improves the dependence on $K$, while the dependence on $T$ increases from constant to $T^{\alpha/2}$. It is worth noting that our algorithm does not rely on the reward gap assumption required by Simchi-Levi and Wang~(2024).
>
> In the adversarial setting without network interference, Liang and Bojinov~(2023) provide an ATE estimation error upper bound of $\tilde{O}(KT^{\alpha - 1/2})$ for $\alpha \in [0, 1/4]$, which matches the performance of our algorithm when specialized to the no-network case. Since their work primarily focuses on ATE inference and does not provide a regret upper bound, a direct comparison of regret upper bounds is not straightforward.
>
> In the stochastic setting with network interference, Zhang and Wang~(2024) propose a two-phase algorithm: the first phase (of length $T_1$) uniformly explores all exposure super arms, while the second phase performs pure regret minimization. They achieve a regret bound of  $R(T, \pi) = \tilde{O}(\sqrt{|\mathcal{U}_E| T} + T_1)$, and an ATE estimation error of $e(T, \hat{\Delta}) = \tilde{O}(\sqrt{|\mathcal{U}_E| / T_1})$, which together yield $\sqrt{R(T, \pi)} \cdot e(T, \hat{\Delta}) = \tilde{O}(\sqrt{|\mathcal{U}_E|})$. While their algorithm is theoretically optimal in this setting, we emphasize two key limitations: (i) its theoretical guarantees hinge on the assumption of a specific reward distribution (i.e., the stochastic bandit setting), and (ii) it lacks support for continual monitoring of ATE inference, which hinders the ability to dynamically balance ATE estimation and regret minimization throughout the learning process. In contrast, our design leverages confidence sequences, which enables continual monitoring of ATE inference and dynamic adjustment of the exploration strategy. Therefore, we claim that our design balances the trade-off between statistical inference and regret minimization, whereas theirs only addresses the trade-off between estimation and regret.
>
> **Reference:**
>
> Waudby-Smith I, Arbour D, Sinha R, et al. Time-uniform central limit theory and asymptotic confidence sequences[J]. The Annals of Statistics, 2024, 52(6): 2613-2640.
>
> Simchi-Levi, David and Wang, Chonghuan, Multi-armed Bandit Experimental Design: Online Decision-making and Adaptive Inference (September 20, 2022). Available at SSRN.
>
> Liang B, Bojinov I. An experimental design for anytime-valid causal inference on multi-armed bandits[J]. arXiv preprint arXiv:2311.05794, 2023.
>
> Zhang Z, Wang Z. Online experimental design with estimation-regret trade-off under network interference[J]. arXiv preprint arXiv:2412.03727, 2024.
>
> Abhineet Agarwal, Anish Agarwal, Lorenzo Masoero, Justin Whitehouse. A Multi-Armed Bandits with Network Interference,
>
> Shishir Adhikari, Sourav Medya, Elena Zheleva. Learning Exposure Mapping Functions for Inferring Heterogeneous Peer Effects

---

> > ### Comment · Reviewer_fNDP · 2025-08-05
> >
> > Thanks for the responses. Please make sure to add them in the paper for completeness and clarity. Also mentioning the limitations and the ablation effect of $\alpha$ would be very helpful to the reader.
> >
> > I have two more questions:
> >
> > 1. It seems from the ablation study that choosing $\alpha$ close to $0.25$ (the legal upper limit) yields a good trade-off. Is it the case? Is it reflected also through the theoretical analysis? Can you explain.
> >
> > 2. As you have mentioned that the proposed framework can work with approximate $H$, the natural question is what is the effect of misspecification/approximation in that case? Do we loose all the theoretical guarantees or even practical efficiency? Is there any easy fix that I am not imagining?

---

> ### Author Response · Authors · 2025-08-06
>
> Thank you for the thoughtful feedback. We will carefully revise the paper to incorporate the discussed points. We provide below our responses to the two questions you brought up.
>
> > 1. It seems from the ablation study that choosing $\alpha$ close to 0.25 (the legal upper limit) yields a good trade-off. Is it the case? Is it also reflected through the theoretical analysis? Can you explain?
>
> We would like to emphasise that the algorithm's ability to achieve a good “trade-off” (in practical terms) depends on the relative importance of regret and ATE estimation error in the specific application. As described in **A2**, if both a small regret and a small ATE estimation error are desired in practice, $\alpha$ can be set close to 0.25, in which case the algorithm can achieve both, as supported by the regret and ATE upper bounds in Theorem 5.6. If the focus is solely on regret minimization, the EXP3 algorithm with network can be used instead, whereas if the sole concern is ATE estimation, $\alpha$ should be set close to 0. In this sense, our algorithm offers a flexible and straightforward approach that can be readily adapted to different scenarios.
>
> If the “trade-off” in question refers to our term $\sqrt{R(\pi,T)} \cdot e(T,\hat{\Delta}) = O_T(T^{\alpha/2})$, the table below (results are calculated from our results in **A6**) clearly shows that as $\alpha$ decreases from 0.25 toward 0, the mean of $\sqrt{R(\pi,T)} \cdot e(T,\hat{\Delta})$ also decreases. This shows that our theoretical results are consistent with our experimental results.
>
> |Parameters    | Mean of  $\sqrt{R(\pi,T)}e(T,\hat{\Delta})$ |
> |-------------|--------------|
> | $\alpha$=0.05| 0.3802 |
> | $\alpha$=0.1 |  0.3797   |
> | $\alpha$=0.15 |  0.3909 |
> | $\alpha$=0.2 |    0.3923   |
> | $\alpha$=0.24| 0.4217   |
>
> > 2. As you have mentioned that the proposed framework can work with an approximate $H$, the natural question is, what is the effect of misspecification/approximation in that case? Do we loose all the theoretical guarantees or even practical efficiency? Is there any easy fix that I am not imagining?
>
> Thank you for your insightful question. When only an approximate version of the adjacency matrix $H$ is observed, our framework allows for a principled adjustment to the analysis. Under standard regularity conditions, the main results remain valid, with the approximation error being properly controlled. We give a brief illustration: Let $\hat{\Delta}_T\left(S_i, S_j\right)$ be our IPW estimator based on the designed exposure super-arms $S_i, S_j \in \mathcal{U}_E$. Denote by $\bar{\tau}_T^{\star}\left(S_i, S_j\right)$ the true ATE computed under the (possibly finer) ground-truth exposure mapping $S^{\star}$. Define the approximation (misspecification) bias $\mathcal{B}(\mathbf{S}, \mathcal{C})$ as the maximium difference between $\bar{\tau}_T^{\star}\left(S_i, S_j\right)$ and $\bar{\tau}_T^{}\left(S_i, S_j\right)$ among all ${S_i, S_j}$. Informally, we prove the biased minimax lower bound under mild assumptions:
>
> $$
> \min _{\pi, \hat{\Delta}} \max _{v \in \mathcal{E}_0} \sqrt{R(T, \pi)} \cdot(e(T, \hat{\Delta})-\mathcal{B}(\mathbf{S}, \mathcal{C}))=\Omega\left(\sqrt{\left|\mathcal{U}_E\right|}\right) .
> $$
>
>
> Intuitively, compared with our Theorem 4.1 in our main text, misspecification bias introduces larger errors. However, we can maintain robustness by controlling the term $e(T, \hat{\Delta})-\mathcal{B}(\mathbf{S}, \mathcal{C})$ to preserve our lower bound to show robustness: even after subtracting the systematic bias, the regret-inference product cannot beat the lower bound barrier; it is a nontrivial addition to the perfectlyspecified setting in the previous literature (also refer to ``Reviewer LDZS``).
>
> For validation of our statement of **Q2**, we could replicate the above experiment in **Q1** with a 0.5%,1%,2%,4%,8% probability of flipping the adjacency matrix $H$. The results demonstrate the relative robustness of our result:
>
> |Parameters    | Mean of  $ \sqrt{R(T, \pi)} \cdot(e(T, \hat{\Delta})-\mathcal{B}(\mathbf{S}, \mathcal{C}))$ |
> |-------------|--------------|
> | |{0.5%,1%,2%,4%,8%}|
> | $\alpha$=0.05| 0.4032, 0.4128, 0.4220, 0.4090, 0.4287|
> | $\alpha$=0.1 | 0.4178, 0.4281, 0.4335, 0.4090, 0.4292 |
> | $\alpha$=0.15 | 0.4198, 0.4307, 0.4371, 0.4532, 0.4437 |
> | $\alpha$=0.2 | 0.4209, 0.4621, 0.4394, 0.4661, 0.4865 |
> | $\alpha$=0.24 | 0.4520, 0.4623, 0.4620, 0.4892, 0.4900 |
>
> We sincerely appreciate your follow-up question and hope that our explanation contributes to a clearer and deeper understanding.

---

> > ### Comment · Reviewer_fNDP · 2025-08-08
> >
> > Thanks for the detailed response. My concerns are satisfactorily addressed and looking forward to the extended draft.

---

> > > ### Author Response · Authors · 2025-08-08
> > > **Thank you for the positive follow-up and feedback.**
> > >
> > > We sincerely appreciate your positive follow-up and feedback. We’ll proceed with revisions in our extended draft.

---

### Official Review · Reviewer_Nu5B · 2025-07-03

**Clarity:** 3
**Significance:** 3
**Originality:** 3
**Rating:** 5
**Confidence:** 3

**Summary:**

The paper establishes theoretical results unifying statistical estimation and regret minimization in the multi-armed bandits with network interference setting. The paper also tests the trade-off with different network topologies and experimental setups.

**Questions:**

1. The paper uses two common techniques to reduce the dimension of the potential outcomes, I wonder what is the role of the clustering assumption beyond just restricting the cardinality of $\mathcal{U}_{\epsilon}$. For example if I only have this exposure mapping but restrict the cardinality of $\mathcal{U}_s$. I think all the results go through?
2. I am a bit confused about the line 357, can you explain why you make that conjecture? Also, can you explain what makes line 339 and (3) so different?
3. Is it possible to add covariates into this setup, for example your potential outcomes depend on the exposure mapping as well as personal characteristics.

**Ethical Concerns:**

["NO or VERY MINOR ethics concerns only"]

**Final Justification:**

All my concerns are resolved and I vote for acceptance.

**Limitations:**

Yes

**Paper Formatting Concerns:**

Noted above about some reference mismatch

**Quality:**

3

**Strengths And Weaknesses:**

Strength:
1. Significance: I think the topic is very interesting to me, unifying different pieces from two topics: statistical estimation and regret minimization. The former is very broadly studied by statistics community and the latter by the online learning community. As shown in table 1, there is no previous work that considers the three objectives in a network interference setup.
2. Quality: The theoretical results are complete and I also like the experiments.
Weaknesses:
1. Clarity:
* Can you make sure the reference works in the main text as well as the appendix? In particular, figure 1 referred to theorem 5.2, 5.3 and 5.4 which are not in the text.
* Some of the notations can be more clear. For example, in the definition of objective 3, the left hand side uses $\nu$ which is a legitimate  instance but the right hand side not.

---

> ### Author Rebuttal · Authors · 2025-07-28
>
> We appreciate the recognition of the importance of our learning problem, as well as the kind words regarding the clarity of our writing. We address the points below.
>
> > ***Q1: Can you make sure the reference works in the main text as well as the appendix? In particular, figure 1 referred to theorem 5.2, 5.3 and 5.4 which are not in the text. Some of the notations can be more clear. For example, in the definition of objective 3, the left hand side uses ν which is a legitimate instance but the right hand side does not.***
>
> ---
>
> **A1:** Thank you for your helpful comments. We will carefully revise the paper to ensure all references are properly defined and linked in both the main text and appendix. We will also clarify notations and correct all typos and grammar issues in the final version.
>
> > ***Q2: The paper uses two common techniques to reduce the dimension of the potential outcomes, I wonder what is the role of the clustering assumption beyond just restricting the cardinality of $\mathcal{U}_E$. For example if I only have this exposure mapping but restrict the cardinality of $\mathcal{U}_s$. I think all the results go through?***
>
> ---
>
> **A2:** Thank you for your question. In our setting, the role of clustering is to restrict the cardinality of the super exposure arm set $U_E$, i.e., $|\mathcal{U}_E| \le d_s^C$, where $C$ denotes the number of clusters. If there is no clustering, this corresponds to the special case $C = N$, where each unit belongs to its own cluster. In that case, we have $|\mathcal{U}_E| \leq d_s^N$, under which all of our theoretical results continue to hold. In fact, the case studied by Agarwal et al., (2024) can be viewed as a setting without clustering, which is a special case of our framework. Please refer to line 192 for a more detailed discussion.
>
> > ***Q3: I am a bit confused about line 357. Can you explain why you make that conjecture? Also, can you explain what makes line 339 and (3) so different?***
>
> ---
>
> **A3:** Thank you for your question. The conjecture in Line 357 was meant to propose a potentially provable lower bound that could match the upper bound achieved by our algorithm. In light of your and Reviewer fNDP’s suggestion, we will remove this conjecture for clarity and instead revise the discussion to focus on explaining why the current upper bound (i.e., Line 339) does not match the existing lower bound (i.e., Eq. (3)).
>
> We believe the gap between the upper and lower bounds can be attributed to inherent structural constraints required by asymptotic confidence sequences (CS). Proposition 2.5 together with Condition L3 of Waudby--Smith et al. (2021) gives a convenient way to construct a valid $(1-\tilde{\delta})$ asymptotic CS in our learning setup: Define
> $\tilde V_t(S_i, S_j)
> = \sum_{s=1}^t \sigma_{s}^2(S_i, S_j)
> = \sum_{s=1}^t \left(\frac{\mathbb{E}[R_s(S_i)^2]}{\pi^{\text{MAD}}s(S_i)}+\frac{ \mathbb{E}[R_s(S_i)^2]}{\pi^{\text{MAD}}s(S_j) }\right)$.
> If there exists $\hat{V}_t(S_i,S_j)$ such that $t^{-1}(\hat V_t-\tilde V_t)(S_i,S_j) \xrightarrow{a.s.}0$, then $\hat{\bar{\tau}}_t(S_i, S_j) \pm \hat C_t(S_i, S_j)$ is a valid ($1-\tilde{\delta}$) asymptotic CS, where $\hat C_t(S_i,S_j)=\sqrt{\frac{2(\hat V_t(S_i,S_j)\eta^2+1)}{t^2\eta^2}
> \log(\frac{\sqrt{\hat V_t(S_i,S_j)\eta^2+1}}{\tilde{\delta}})}$.
>
> To ensure $t^{-1}(\hat V_t-\tilde V_t)(S_i,S_j) \xrightarrow{a.s.}0$, we construct $\hat V_t(S_i,S_j)$ via the unbiased IPW estimator that is fully computable from the data: $\hat V_t(S_i, S_j) = \sum_{s=1}^t  \frac{ \mathbf{1}( S_{s} = S_i ) \cdot R_{s}(S_i)^2 }{ ( \pi^{\text{MAD}}s(S_i) )^2 }  + \frac{ \mathbf{1}( S_{s} = S_j ) \cdot R_{s}(S_j)^2 }{ ( \pi^{\text{MAD}}s(S_j) )^2 }$. Let $M_s(S_i,S_j):=(\hat \sigma_s^2- \sigma_s^2)(S_i,S_j)$. Then $M_s(S_i,S_j)$ is a martingale difference sequence and $t^{-1}\sum_{s=1}^t M_s(S_i,S_j)= t^{-1}(\hat V_t-\tilde V_t)(S_i,S_j)$. By the SLLN for martingale difference sequence, if $\sum_{t=1}^\infty \mathbb{E}[M_t^2(S_i,S_j)]/t^2<\infty$, we obtain $t^{-1}(\hat V_t-\tilde V_t)(S_i,S_j)\xrightarrow{a.s.}0$. This requires $(\hat \sigma_t^2 - \sigma_t^2)^2(S_i,S_j) = o(t)$ (by a $p$-series test).
> In our design this summability is guaranteed by $1/\pi_t^{\mathrm{MAD}}(S)=o(t^{1/4})$ for all $S\in \mathcal{U}_E$. Hence, when $\pi_t^{\mathrm{MAD}}(S)\propto t^{-\alpha}$, we require $\alpha\in[0,1/4)$. Under this requirement the CS width behaves as $\hat C_t=\tilde{O}(t^{\alpha-1/2})\quad \text{for }\alpha\in[0,1/4)$,
> and enforcing $1/\pi_t^{\mathrm{MAD}}(S)=o(t^{\alpha})$ for all $S\in\mathcal{U}_E$ induces additional regret $\tilde{O}(t^{1-\alpha})$. This ultimately leads to the gap between the lower and upper bounds.
>
> > ***Q4: Is it possible to add covariates into this setup, for example your potential outcomes depend on the exposure mapping as well as personal characteristics.***
>
> ---
>
> **A4:** Thank you for the insightful question. We believe that the covariate-assisted exposure modeling framework (offline setting) studied by Leung et al. (2022) can be naturally extended to our online learning setting. In their work, graph neural networks (GNNs) are employed to estimate the ATE under network interference. Meanwhile, several studies in the bandit literature, such as Kassraie et al. (2022), have explored the integration of GNNs with structured decision-making. These advances indicate that applying GNN-based algorithms and estimators to our generalized setting (i.e., our setting with covariates) is both natural and technically feasible. We leave this interesting direction for future investigation.
>
> **Reference:**
>
> Leung M P, Loupos P. Graph neural networks for causal inference under network confounding[J]. arXiv preprint arXiv:2211.07823, 2022.
>
> Kassraie P, Krause A, Bogunovic I. Graph neural network bandits[J]. Advances in Neural Information Processing Systems, 2022, 35: 34519-34531.
>
> Waudby-Smith I, Arbour D, Sinha R, et al. Time-uniform central limit theory and asymptotic confidence sequences[J]. The Annals of Statistics, 2024, 52(6): 2613-2640.
>
> Agarwal A, Agarwal A, Masoero L, et al. Multi-Armed Bandits with Network Interference[J]. Advances in Neural Information Processing Systems, 2024, 37: 36414-36437.

---

> > ### Comment · Reviewer_Nu5B · 2025-08-02
> > **Response to rebuttal**
> >
> > Thanks for your response! It clears my confusions and concerns.

---

> ### Author Response · Authors · 2025-08-06
> **Thank you for your kind feedback**
>
> Many thanks for your kind feedback — we're glad the clarification was helpful.

---

### Decision · Program_Chairs · 2025-09-17

**Decision:**

Accept (poster)

**Comment:**

This paper studies the problem of bandits with network interference under objectives of jointly minimizing the regret and estimating the treatment effect in the adversarial environment. The paper establishes an essential tradeoff between them and constructs an algorithm, EXP3-N-CS that tries to optimize it. Though the derived lower and upper bounds have some gap, detailed experiments are conducted to examine the tradeoff.

The reviewers agreed in the opinions that the paper is well-motivated and has solid technical contributions. While several questions and concerns such as the ones on the presentation and practical aspects are raised, they are well-addressed in the rebuttal. Based on these observations I determined to recommend acceptance. I would expect that the authors polish the paper in the final version by incorporating the discussions with the reviewers.

While this paper makes good contribution in the area of causal inference, my concern is the interest to general NeurIPS community. Indeed, while the high-level message on the tradeoff between the regret minimization and the treatment effect estimation is very interesting, the explanation of the model heavily relies on the context of causal inference (in particular, seemingly [Leung 2022a]) and it is very hard to understand. For example, the meaning of "exposure" is explained nowhere despite its heavy use. Combined with the inappropriate or wrong notation (please see below), I gave up to understand the formal definition of the considered model, which seems to also apply to other readers outside the area of causal inference. I would expect that the authors authors also address this point so that the paper becomes of broader interest.

Examples of the problems I encountered when reading the model is as follows.
- Since no intuitive meaning of "exposure mapping" is introduced, the exposure mapping $\mathbf{S}$ has to be just regarded as an abstract function in the form of (1). Here, the second $\mathbb{H}$ in (1) seems to be a typo, since it must be a domain of the first $\mathbb{H}$. $\mathcal{U}_s$ is just explained as "the set with $d_s$ exposure arm", but since (1) is the equation to define the "exposure arm", this explanation is cyclic. Similarly, it is strange that the notation like $d_s$ appears in the definition of $s$.
- In addition, the example in the L176 says that $\mathcal{U}_c$ is not a set and is 0 or 1, which is incompatible with the description of (1). Is this a typo for $\mathcal{U}_s=\\{0,1\\}$? I don't see why in this case $\mathcal{U}_C$ is automatically determined without specifying the function $\mathbf{S}$.
- $S_t$ is undefined. I guess it is $\mathbf{S}(i, A_t, \mathbb{H})$ but, if so, what is $\mathbb{P}(S_t|\mathcal{H}_t)$ in (2)? Is this a random variable such that $\mathbb{P}(S_t=S|\mathcal{H}_t)$ where $S_t$ (a random variable) is substituted to $S$? Does this mean something like $\pi_t(A, S_t)$ rather than $\pi_t(A)$?
- I guess that $\mathbf{\mathrm{S}}$ in (1) is a typo for the italic. Similarly there also seems to be a confusion for $C_q$ and $\mathcal{C}_q$.

In general, since the notion of the exposure mapping seems to be a part of the model itself, I believe that the formal definition and its advantage should not be described in a mixed way like a current one.